# Multi-compartmental model of glymphatic clearance of solutes in brain tissue

**Alexandre Poulain[1,3]\*, Jørgen Riseth[2,3], Vegard Vinje[3]**

**1** Laboratoire Paul Painlevé, UMR 8524 CNRS, Université de Lille, Lille, France, **2** Department of Mathematics, University of Oslo, Oslo, Norway, **3** Department for Numerical Analysis and Scientific Computing, Simula Research Laboratory, Oslo, Norway

\* alexandre.poulain@univ-lille.fr

**Data Availability Statement:** The computational code and meshes used to generate the results in this study are freely available at GitHub: https://github.com/jorgenriseth/multicompartment-solute-transport.

## Abstract

The glymphatic system is the subject of numerous pieces of research in biology. Mathematical modelling plays a considerable role in this field since it can indicate the possible physical effects of this system and validate the biologists' hypotheses. The available mathematical models that describe the system at the scale of the brain (*i.e.* the macroscopic scale) are often solely based on the diffusion equation and do not consider the fine structures formed by the perivascular spaces. We therefore propose a mathematical model representing the time and space evolution of a mixture flowing through multiple compartments of the brain. We adopt a macroscopic point of view in which the compartments are all present at any point in space. The equations system is composed of two coupled equations for each compartment: One equation for the pressure of a fluid and one for the mass concentration of a solute. The fluid and solute can move from one compartment to another according to certain membrane conditions modelled by transfer functions. We propose to apply this new modelling framework to the clearance of $^{14}$C-inulin from the rat brain.

## 1 Introduction

The proposed glymphatic system [1] explains clearance of metabolic waste from the brain and has been the subject of many pieces of research in the past decade [1–4]. The glymphatic theory suggests that clearance of metabolic solutes in the brain is facilitated by specific pathways for exchange between interstitial fluid (ISF) and cerebrospinal fluid (CSF). This exchange occurs via perivascular spaces (PVSs), small fluid-filled spaces surrounding blood vessels. According to the glymphatic theory, CSF enters the parenchyma via periarterial spaces and exits it via perivenous spaces. Furthermore, Iliff *et al.* [1] suggested that a bulk flow of fluid occurs in the interstitial space between periarterial and perivenous spaces draining metabolic waste out of the brain. Understanding the glymphatic system is critically important since its impairment may be linked to neurodegenerative diseases such as Alzheimer's disease [5].

Even after a decade of research to verify this theory, many questions remain to be answered: *i)* Does the circulation of CSF as described by Iliff *et al.* [1] (inflow around arteries and outflow around veins) occur? *ii)* What are the mechanisms explaining the movement of CSF in the

**Funding:** The author(s) received no specific funding for this work.

**Competing interests:** The authors have declared that no competing interests exist.

perivascular spaces? *iii)* Does convection in the interstitial space occur, and is this flow sufficient to dominate transport?

In vivo studies using two-photon microscopy have imaged flow along periarterial spaces at the pial surface in the same direction as blood [6, 7], suggesting these spaces act as an entry to the brain. However, the direction and magnitude of flow in penetrating PVS is still debated [8]. Furthermore, the question of the existence of a bulk flow of fluid within the extracellular space (ECS) as proposed by Iliff *et al.* [1] remains open. Indeed, some pieces of research indicate that solute transport in the ECS is dominated by diffusion [2, 9, 10], while others claim that diffusion alone can not explain the transport of tracer within the brain [11–13]. In a recent study, Ray *et al.* [14] concluded that the transport of large molecules is dominated by convection, given the expected ECS flow rates reported in the literature. Convection-diffusion equations have been widely used to study transport within the brain [2, 11, 12, 14–17]. These works helped to gain some insights into the relevant mechanisms that may play a role in the clearance of interstitial solutes. However, in these works, the fluid velocities and concentrations are averaged between ECS and PVS (and all other routes of transport) to capture the overall spread of solutes. For a review of current fluid models available for mathematical and computational representations of the glymphatic system, we refer the reader to [18].

In contrast, coupled discrete-continuous models that can represent different structures such as blood vessels and tissue have been used to study, for example, drug transport to the lung [19], the brain [20] and tissue in general [21] with great detail. However, these models require detailed information on vessel structure and require too many degrees of freedom to study the brain at the macroscale.

To circumvent these limitations, homogenized models have been successfully applied to represent infiltration in porous media [22]. Such framework has been successfully applied to represent the transport of solute and fluid in the ECS and vascular network of vascularized tumors [23–25]. In full-scale patient-specific geometries, multiple-network poroelastic theory (MPET) has been used to study exchange between multiple fluid compartments contained within the (elastic) brain tissue [26–32]. However, the MPET equations have not yet been investigated in terms of the transport of tracers or solutes in the context of the glymphatic system.

In this paper, we therefore develop a homogenized model to describe the glymphatic system and the blood flow at the scale of the rat brain (Fig 1). To validate the relevancy of our modelling framework, we study the clearance of $^{14}$C-inulin from the rat brain. $^{14}$C-inulin is known not to cross the blood-brain-barrier (BBB) and is therefore well suited for investigating the mechanisms behind the clearance of large proteins from the brain tissue. Consequently, it has been used in experimental studies of the glymphatic system (e.g. [33, 34]), which provide reference values for the expected clearance rates of our model. In particular, the presented multi-compartment model represents a movement of CSF through different structures, including the PVSs, the ECS and the vasculature, while the subarachnoid space (SAS) is included via boundary conditions. This modelling of the fluid movement is coupled to a diffusion-convection equation for each compartment to represent the clearance of $^{14}$C-inulin from the brain. Our model suggests that without blood filtration, transport is explained mainly by diffusion within the brain. However, when ISF is allowed to filtrate across the vascular wall, PVS flow is reversed, and clearance from the ECS is substantially increased.

## 2 Methods

### 2.1 Mathematical models

**2.1.1 Notations.** We denote by $\Omega \subset \mathbb{R}^3$ the spatial domain, *i.e.* the rat brain. We assume that the boundary $\partial\Omega$ of this domain is sufficiently smooth. Therefore, we denote by $\mathbf{x} \in \Omega$ any

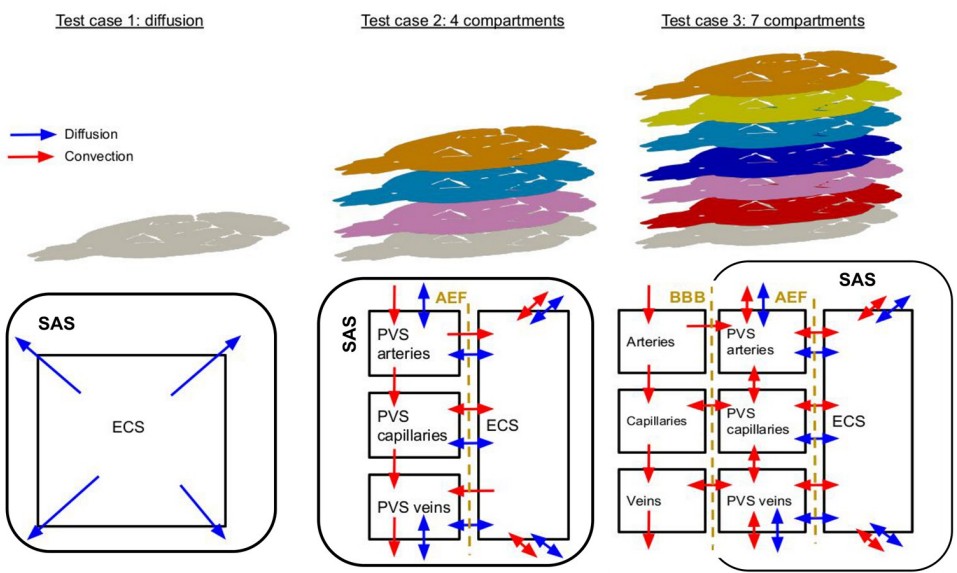

**Fig 1. Illustrative representation of the three test cases.** Red arrows indicate the movement of fluids through the compartments and blue arrows the diffusive movement of [14]C-inulin. Double arrows indicate that the movement could be directed in both directions and is, a priori, not known. With exception of the blood compartments, the arrows pointing to the outside of any compartment denotes a connection of this compartment with the subarachnoid space. *AEF* denotes the *astrocyte endfeet barrier* and *BBB* the *blood-brain barrier*.

point of this domain such that the coordinates are given by $\mathbf{x} = (x_1, x_2, x_3)$. Bold symbols will be used to denote vectors. Since we model the time evolution of the glymphatic system, our time-space domain is denoted by $\Omega_T = \Omega \times [0, T]$ for some finite $T > 0$. We test two mathematical models: A pure diffusion model in a single compartment and a multi-compartment model, which includes both diffusive and convective transport. When an unknown or a parameter is indicated with a subscript, it denotes its compartment. The subscripts $a$, $c$, and $v$ denote the arterial, capillary and venous blood networks, respectively. Similarly, the subscripts $pa$, $pc$, $pv$ denote the periarterial, pericapillary and perivenous fluid networks. The subscript $e$ indicates the ECS.

**2.1.2 The diffusion equation.** Denoting by $c_e = c_e(t, \mathbf{x})$ the solute concentration in ISF, the diffusion equation reads

$$\frac{\partial c_e}{\partial t} = D_e^* \Delta c_e, \quad \forall \mathbf{x} \in \Omega, \quad t \in (0, T]. \tag{1}$$

Here, $D_e^*$ is the effective diffusion coefficient of [14]C-inulin in the ECS and $\Delta = \nabla \cdot \nabla$ is the Laplace operator *i.e.* $\Delta f = \frac{\partial^2 f}{\partial x_1^2} + \frac{\partial^2 f}{\partial x_2^2} + \frac{\partial^2 f}{\partial x_3^2}$ (for a general scalar function $f$).

**2.1.3 The multi-compartment model.** To take into account the different structures in which the fluid flows, we consider the multiple compartments as depicted in the schematic illustrations given in Fig 1. We denote by $J$ the set of compartments ($J$ can thus be modified to describe all three test cases shown in Fig 1), and we denote the pressure in the $j$–th compartment by $p_j = p_j(t, \mathbf{x})$ and for the solute concentration, $c_j = c_j(t, \mathbf{x})$. The fluid flow in our model is computed via static MPET equations [28, 29] without displacements. The velocity fields are defined using Darcy's law [35] for flow in porous media, which stipulates that the velocity is

proportional to the opposite of the gradient of the fluid's pressure, *i.e.*

$$\mathbf{v}_j = -\frac{\kappa_j}{\mu_j}\nabla p_j, \tag{2}$$

where $\kappa_j$ is the permeability coefficient for the fluid in compartment $j$ and $\mu_j$ is the dynamic viscosity of the fluid in the compartment. We denote by $\phi_j$ the porosity of the $j$−th compartment (*i.e.* the relative volume taken by the pores of this compartment). We emphasize that the compartments are all present at any point $\mathbf{x} \in \Omega$. Under the assumption of incompressible flow, then for all $\mathbf{x} \in \Omega$, $t \in (0, T]$, the equations' systems for each $j \in J$ is given by

$$\begin{cases} -\nabla \cdot \left(\frac{\kappa_j}{\phi_j\mu_j}\nabla p_j\right) = r_j, \\ \frac{\partial c_j}{\partial t} - \frac{\kappa_j}{\phi_j\mu_j}\nabla \cdot \left(c_j\nabla p_j\right) - D_j^*\Delta c_j = s_j. \end{cases} \tag{3}$$

Here, $D_j^*$ is the effective diffusion coefficient in the $j$-th compartment, and $r_j$, $s_j$ are the transfer functions to model the exchanges between the compartments and will be described in the next paragraph.

**Remark 1** *For simplicity reasons, we consider the porosity, permeability and diffusion coefficients to be homogeneous, i.e. no spatial variation is considered for these parameters.*

**Remark 2** *We note that $c_j$ denotes the microscopic fluid concentration, which is related to the macroscopic or total concentration via $c_j^{total} = c_j * \phi_j$.*

**2.1.4 Transfer functions.** The transfer functions in System (3) model the exchange of fluid, $r_j$, and solutes, $s_j$, between the different compartments. These compartments are either separated by a membrane or directly connected to vessels along the same tree (*e.g.* an artery branching to capillaries or the PVS around arteries branching to the PVS around capillaries. We assume the possibility of PVS around capillaries in line with *e.g.* [36]).

When the compartments are separated by a membrane, the fluid flows from one compartment to another due to a difference in pressure which is related to the hydraulic conductivity of the membrane, *i.e.*

$$r_j = \frac{1}{\phi_j}\sum_{i \in J, i \neq j}\gamma_{j,i}\left[(p_i - p_j) - \sigma_{i,j}(\pi_i - \pi_j)\right], \tag{4}$$

with

$$\gamma_{j,i} = L_{i,j}\frac{|S_{i,j}|}{|\Omega|}, \tag{5}$$

where $|\Omega| = \int_\Omega 1 \, d\mathbf{x} = 2313 \text{ mm}^3$ is the brain volume (computed from our rat brain mesh), $L_{i,j}$ is the hydraulic conductivity of the membrane separating the $i$−th and $j$−th compartments, $\frac{|S_{i,j}|}{|\Omega|}$ is the ratio between the surface of the membrane and the total volume of the tissue, and $\sigma_{i,j}$ is the osmotic reflection coefficient for the membrane. This reflection coefficient corresponds to a specific solute. In this work, we only consider osmotic effects due to plasma cells in the blood where $\pi_j$ is the osmotic pressure. The solute crosses the membrane due to the combination of two effects: Either via convection of fluid through the pores of the membrane or via diffusion.

These two effects are modelled by the transfer functions (see *e.g.* [37])

$$s_j = \frac{1}{\phi_j} \sum_{i \in J, i \neq j} \lambda_{j,i}(c_i - c_j) + \frac{(c_j + c_i)}{2} \tilde{\gamma}_{j,i}(p_i - p_j - \sigma_{i,j}(\pi_i - \pi_j)),$$ (6)

where this time

$$\lambda_{j,i} = P_{i,j} \frac{|S_{i,j}|}{|\Omega|}, \quad \tilde{\gamma}_{j,i} = \gamma_{j,i}(1 - \sigma_{\text{reflect}}),$$

in which $P_{i,j}$ is the permeability of the membrane separating the $i$–th and $j$–th compartments to the solute and $\sigma_{\text{reflect}}$ reflects the solvent-drag reflection coefficient.

In the case of a continuous transition between compartments, such as between arteries and capillaries, no membrane is present and we set $P_{i,j} = 0$. Values for the exchange coefficients $\gamma_{j,i}, \tilde{\gamma}_{j,i}$ and $\lambda_{j,i}$ are given in Subsection 2.3.

**2.1.5 Clearance of $^{14}$C-inulin.** To study the clearance of $^{14}$C-inulin from the rat brain, we consider 3 model variations.

We first assume that the bulk flow of fluid in the interstitial space is negligible and transport occurs only due to diffusion in the interstitial space. Hence, we use Eq (1). Clearance of $^{14}$C-inulin occurs at the brain surface and is modelled by appropriate boundary conditions described below. This scenario is represented by Test case 1 on Fig 1.

Secondly, we consider a clearance of $^{14}$C-inulin due to the glymphatic system. Hence, we use System (3) with $|J| = 4$ compartments: ECS, PVS around arteries, PVS around capillaries, and PVS around veins. Test case 2 in Fig 1 depicts this scenario. CSF is assumed to flow from the PVS around arteries to the PVS around capillaries or in the ECS. From the PVS around capillaries, CSF flows to the ECS or the PVS around veins. From the ECS, CSF may be reabsorbed in the PVS around veins or capillaries. Clearance from the brain may occur at the brain surface from the ECS, the PVS around veins and the PVS around arteries.

Thirdly, we add the effect of blood vasculature. Indeed, cerebral blood vessels are not impermeable, and some fluid could leak from them to the other structures [38]. This case is depicted by Test case 3 in Fig 1.

For the sake of clarity, in the following, we refer to these 3 applications of our modelling framework as

- **Pure diffusion model: Test case 1.** Diffusion only in the interstitial space modeled by Eq (1).

- **4-compartment model: Test case 2.** Clearance from the glymphatic system using System (3) with $J = 4$ compartments.

- **7-compartment model: Test case 3.** Clearance from the glymphatic system and considering the blood perfusion that could affect fluid movement using System (3) with $J = 7$ compartments.

## 2.2 Initial and boundary conditions

**2.2.1 Initial condition.** We consider the application in which the solute is injected directly into the ECS of the rat brain and assume that the initial $^{14}$C-inulin concentration is given as a three-dimensional Gaussian around the centre of injection **s** (see Fig 2b),

$$c_e(0, \mathbf{x}) = C^0 \exp -\frac{|\mathbf{x} - \mathbf{s}|^2}{\sigma^2}$$ (7)

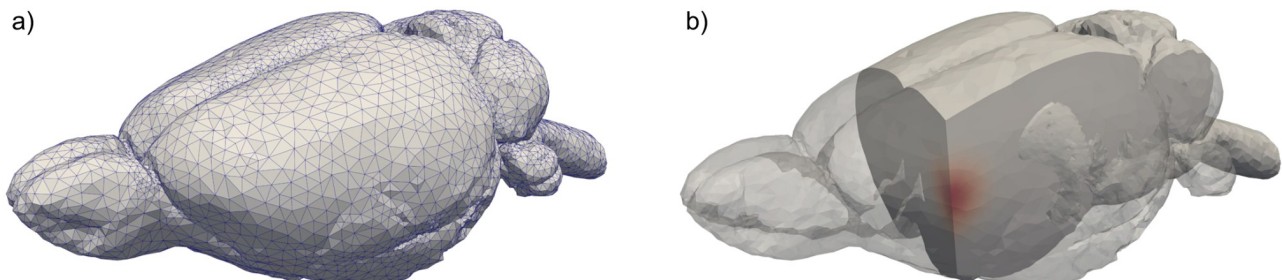

**Fig 2.** a): The computational mesh of the rat brain used for most of the simulations within this article. The meshing procedure is described in section 2.6. For the given mesh, the maximum cell size is $\approx 1/32$ times the diameter of the mesh. b): The initial $^{14}$C-inulin concentration within the ECS, simulating an injection directly into the brain tissue.

where $C^0$ is a reference concentration, and $\sigma$ determines the initial spread of the solute after injection. The reference concentration is chosen such that the integral of the initial condition over the domain matches the injected tracer amount. Already for $\sigma = 1$mm, the initial condition quickly decays towards zero. More than 96% of the total mass is located within a ball of radius 2mm surrounding the injection point, and the concentration values outside of this region have minimal impact. However, to ensure that also the initial condition adheres to the prescribed boundary conditions (below), we project the initial condition onto a space of functions which vanish at the boundary. We emphasize that the initial condition is the same for the case of a single compartment and when multiple compartments are considered. In the following, we present numerical results for which the initial point of injection is located in the right hemisphere with coordinates $\mathbf{s} = (4, 2, 3)$.

**2.2.2 Boundary conditions.** To generate a relevant bulk flow within the PVSs, we assume a slight pressure difference between the boundary of the PVSs around arteries and veins. We know that intracranial pressure in a rat is $4\pm0.74$mmHg (see [39]). ISF pressure has been measured in rat [40] and is $3.43\pm0.65$mmHg.

Therefore, we supplement the pressure equations with

$$\begin{cases} -\frac{\kappa_e}{\mu_{\text{CSF}}} \frac{\partial p_e}{\partial \mathbf{v}}(t, \mathbf{x}) = L_{e,SAS}(p_{SAS} - p_e), & -\frac{\kappa_{pa}}{\mu_{\text{CSF}}} \frac{\partial p_{pa}}{\partial \mathbf{v}}(t, \mathbf{x}) = L_{\text{PVSpial},pa}(p_{\text{PVSpial}} - p_{pa}), \\ \frac{\partial p_{pc}}{\partial \mathbf{v}}(t, \mathbf{x}) = 0, & p_{pv} = 3.26 \, \text{mmHg}, \end{cases} \tag{8}$$

on $\partial\Omega$, $t > 0$, with $\mathbf{v}$ being the outward-pointing normal vector to the boundary $\partial\Omega$, $p_{\text{PVSpial}} = 4.74$mmHg the CSF pressure inside the PVS of pial arteries and $p_{SAS} = 3.26$mmHg is the CSF pressure inside the SAS. We emphasize that these pressure values have been chosen to be in the biologically relevant threshold compared to measurements [39]. The coefficients $L_{\text{PVSpial},pa}$ and $L_{SAS,e}$ are related to the permeability of the pial surface of the brain for the CSF (specified in S1 Appendix).

If cerebral blood perfusion is included in the model(test case 3), then fluid movement is affected and we need additional parameters, namely

$$\begin{cases} -\frac{\kappa_a}{\mu_a} \frac{\partial p_a}{\partial \mathbf{v}}(t, \mathbf{x}) = \frac{B_{\text{blood}}}{|\partial\Omega|} \\ \frac{\partial p_c}{\partial \mathbf{v}}(t, \mathbf{x}) = 0, \quad p_v(t, \mathbf{x}) = 7.0 \, \text{mmHg}, \end{cases} \quad \text{on } \partial\Omega, \, t \geq 0, \tag{9}$$

with $B_{\text{blood}} = 2.32$ mL/min (see Table 7 in S1 Appendix and assuming a 2g rat brain) and $|\partial\Omega|$ is the area of the surface of the rat brain.

For the concentration equations, different boundary conditions are considered. The first and simplest approach is to use homogeneous Dirichlet boundary conditions to represent clearance from the tissue and zero-flux boundary conditions for the compartments that are not in communication with the SAS. Since the periarterial, perivenous and extracellular spaces represent possible outflow routes, we impose Dirichlet boundary conditions for the concentration equations in these compartments. For the other compartments, we assume that there is no flow at the brain's surface. Thus, we have

$$
\begin{cases}
c_j|_{\partial\Omega} = 0, \quad \text{for} \quad j = \{pa, pv, e\}, \\[2mm]
\dfrac{\partial \left( D_j \nabla c_j + \frac{\kappa_j}{\mu_j} c_j \nabla p_j \right)}{\partial v} = 0 \quad \text{on } \partial\Omega, \text{ and for } j = \{pc\}.
\end{cases}
$$

This condition assumes that no membrane restricts $^{14}$C-inulin movement over the pial surface. Moreover, the clearance of solutes from the SAS is assumed to be sufficiently quick so that $^{14}$C-inulin concentration in the CSF stays zero.

Alternatively, the solute concentration in the CSF within the SAS may be represented by a time-dependent boundary condition. Still assuming instant absorption at the surface, we modify the Dirichlet boundary conditions to

$$
c_j|_{\partial\Omega} = g(t), \quad \text{for} \quad j = \{pa, pv, e\}, \quad t > 0, \tag{10}
$$

where $g(t)$ is given as the total amount of $^{14}$C-inulin that has been cleared from the brain up to that time, averaged over the CSF volume $V_{\text{CSF}}$ in the fluid-filled space surrounding the brain, *i.e.* the SAS. The rate of change of $^{14}$C-inulin tracer within the brain per unit of time is given by

$$
\frac{d}{dt} \int_\Omega \sum_{j \in J} \phi_j c_j \, d\mathbf{x} = \sum_{j \in J} \int_\Omega \phi_j \frac{\partial c_j}{\partial t} \, d\mathbf{x} = -\int_{\partial\Omega} \mathbf{q} \cdot v \, ds, \tag{11}
$$

in which $\mathbf{q}$ is the total mass flux from all the compartments at the surface of the brain (we recall that $v$ is the outward pointing normal to the surface of the brain). For each compartment, this flux is given by the combination of diffusion and convection

$$
\mathbf{q} = \sum_{j \in J} - D_j \nabla c_j + c_j \mathbf{v}_j, \quad \mathbf{v}_j = -\frac{\kappa_j}{\mu_j} \nabla p_j.
$$

A decrease of $^{14}$C-inulin within the brain corresponds to an increase of concentration in the SAS, and vice-versa. Therefore, $g$ satisfies the linear ordinary differential equation

$$
\begin{cases}
\dfrac{dg}{dt} = -\alpha g(t) + \dfrac{1}{V_{\text{CSF}}} \int_{\partial\Omega} \mathbf{q} \cdot v \, ds, \\[3mm]
g(0) = 0,
\end{cases} \tag{12}
$$

where $\alpha > 0$ is the rate of CSF absorption from the SAS. This model assumes instantaneous absorption of $^{14}$C-inulin in the CSF and instant mixing of the solute within the whole SAS.

If $\alpha = 0$, the latter Dirichlet boundary condition may be interpreted as a model for conservation of intracranial $^{14}$C-inulin. Assuming that $^{14}$C-inulin is not eliminated from the SAS, an alternate formulation of this condition is given by

$$
\sum_{j \in J} \int_\Omega \phi_j c_j \, dx + g(t) V_{\text{CSF}} = N_0, \tag{13}
$$

where $N_0 = \sum_{j \in J} \int_\Omega \phi_j c_j(0, \mathbf{x}) \, dx$ is the total amount of $^{14}$C-inulin initially injected into the brain. Thus, for this case, $g$ is given by

$$g(t) = \frac{1}{V_{\text{CSF}}} \left( N_0 - \sum_{j \in J} \int_\Omega \phi_j c_j \, d\mathbf{x} \right). \tag{14}$$

We test the effect of all three different concentration boundary conditions (Homogeneous, Conservation (10) with Eq (14), and Decay (10) with Eq (12)) on clearance of $^{14}$C-inulin from the brain.

## 2.3 Parameter values

**2.3.1 For the convection-diffusion equation.** *2.3.1.1 $^{14}$C-inulin diffusion coefficient.* The free diffusion coefficient for $^{14}$C-inulin is $D_{\text{free}} = 2.98 \times 10^{-4}$mm$^2$/s as reported in [41], and the tortuosity of the rat brain is given by $\lambda = 1.7$ (see [42]). Hence, the effective diffusion coefficient of $^{14}$C-inulin in the rat brain is given by

$$D^* = \frac{D_{\text{free}}}{\lambda^2} = 1.03 \times 10^{-4} \text{ mm}^2/\text{s}.$$

**2.3.2 For the multi-compartment model.** *2.3.1.2 Porosity coefficients.* From [43], we know that the volume fraction of the extracellular space of rats is

$$\phi_e = 0.14.$$

From [44], the volume fraction of blood is estimated to be

$$V_{\text{Blood}} = 3.29 \times V_{\text{Brain}}/100.$$

Furthermore, using the fractions of arteries, veins, and capillaries stated in [45], we obtain

$$\phi_a = 0.0069, \quad \phi_c = 0.011, \quad \phi_v = 0.015.$$

The porosity of the PVS in human white matter is estimated to be around 1% [46]. This value is unknown for the rat. Hence, we assume that the relation holds without relying on measurements. Based on percentages of arterial, venous, and capillary blood volume, we assume a similar volume fraction distribution for the corresponding perivascular spaces:

$$\phi_{pa} = 0.0021, \quad \phi_{pc} = 0.0033, \quad \phi_{pv} = 0.0046.$$

*2.3.1.3 Fluid parameters.* The interstitial fluid and plasma in the blood compartments are assumed to possess different properties. The dynamic viscosity of blood and CSF is given by respectively [32] and [47]. We have

$$\mu_a = \mu_v = \mu_c = 2.67 \times 10^{-3} \text{ Pa s}, \text{ and } \mu_{pa} = \mu_{pv} = \mu_{pc} = \mu_e = 7.0 \times 10^{-4} \text{ Pa s}.$$

In [48], the authors used experimentally obtained resistance coefficients for several compartments. From the definition of these resistances, we can compute the permeability coefficients in several compartments (see S1 Appendix for details). The remaining permeabilities

**Table 1. Baseline fluids (Blood and CSF) viscosity, permeability, porosity and diffusion parameters.**

| Symbol | Unit | Meaning | Value | Reference |
|--------|------|---------|-------|-----------|
| $D$ | mm$^2$/s | Free diffusion coefficient | $D_{\text{free}}^{14\text{C}-\text{inulin}} = 2.98 \times 10^{-4}$ | [41] |
| $D^*$ | mm$^2$/s | Apparent diffusion coefficient | $D^{*,14\text{C}-\text{inulin}} = 1.03 \times 10^{-4}$ | [41] |
| $\kappa_j$ | mm 2 | Permeability | $\kappa_a = 3.30 \times 10^{-6}, \kappa_v = 6.59 \times 10^{-6}, \kappa_c = 8.8 \times 10^{-9},$ | [2] and computed |
| | | | $\kappa_{pa} = 1.0 \times 10^{-11}, \kappa_{pv} = 6.51 \times 10^{-9}, \kappa_{pc} = 3.54 \times 10^{-13}, \kappa_e = 2.0 \times 10^{-11}$ | |
| $\phi_j$ | No unit | Porosity | $\phi_e = 0.14, \phi_a = 0.0071, \phi_c = 0.011, \phi_v = 0.016$ | [43, 44, 51] |
| | | | $\phi_{pa} = 0.0021, \phi_{pc} = 0.0033, \phi_{pv} = 0.0046$ | and computed |
| $\mu_j$ | | Viscosity | $\mu_{pa} = \mu_{pv} = \mu_{pc} = \mu_e = 7.0 \times 10^{-4}$ | [47] |
| | | | $\mu_a = \mu_v = \mu_c = 2.67 \times 10^{-3}$ | [29] |

are obtained from [49, 50].

$$\kappa_a = 3.30 \times 10^{-6} \text{ mm}^2, \quad \kappa_v = 6.59 \times 10^{-6} \text{ mm}^2, \quad \kappa_c = 8.8 \times 10^{-9} \text{ mm}^2,$$

$$\kappa_{pa} = 1.0 \times 10^{-11} \text{ mm}^2, \quad \kappa_{pv} = 6.51 \times 10^{-9} \text{ mm}^2, \quad \kappa_{pc} = 3.54 \times 10^{-13} \text{ mm}^2,$$

$$\kappa_e = 2.0 \times 10^{-11} \text{ mm}^2.$$

The baseline values for the fluid parameters are summarized in Table 1.

*2.3.1.3 Exchange coefficients.* We start with the exchange coefficients from blood to tissue, *i.e.* $\gamma_{e,a}$, $\gamma_{e,c}$, $\gamma_{e,v}$, defined by

$$\gamma_{j,i} = L_{i,j} \frac{|S_{i,j}|}{|\Omega|}.$$

As in [52], we use the hydraulic conductivities reported in [53–55]. We use the following values

$$L_{a,e} = 9.1 \times 10^{-10} \text{ mm}/(\text{s Pa}), \quad L_{c,e} = 1.0 \times 10^{-10} \text{ mm}/(\text{s Pa}), \quad L_{v,e} = 2.0 \times 10^{-11} \text{ mm}/(\text{s Pa}).$$

Furthermore, from [56], we estimate the ratio between the surface area of capillaries and brain volume to

$$\frac{|S_{a,e}|}{|\Omega|} = 9 \text{ mm}^{-1}.$$

Using the computations performed in [57], we assume that the surface density of capillaries is three times greater than the surface density of arteries and veins, *i.e.*

$$\frac{|S_{a,e}|}{|\Omega|} = 3 \text{ mm}^{-1}, \quad \frac{|S_{v,e}|}{|\Omega|} = 3 \text{ mm}^{-1}.$$

Altogether, we obtain

$$\gamma_{e,a} = 2.7 \times 10^{-9} \text{ (s Pa)}^{-1}, \quad \gamma_{e,c} = 9.0 \times 10^{-10} \text{ (s Pa)}^{-1}, \quad \gamma_{e,v} = 6.0 \times 10^{-11} \text{ (s Pa)}^{-1}.$$

Then, we turn to the values of the exchange parameters from PVSs to ECS, *i.e.* $\gamma_{e,pa}$, $\gamma_{e,pc}$, $\gamma_{e,pv}$. From the 1D resistance parameters in [48], we compute the following coefficients

(see S1 Appendix for details about the computations)

$$\gamma_{e,pa} = 2.2 \times 10^{-7} \ (\text{s Pa})^{-1}, \quad \gamma_{e,pc} = 9.2 \times 10^{-9} \ (\text{s Pa})^{-1}, \quad \gamma_{e,pv} = 2.0 \times 10^{-7} \ (\text{s Pa})^{-1}.$$

From the previous values, we determine the following exchange coefficients for the transfer between blood vessels and PVSs (see S1 Appendix for details)

$$\gamma_{pa,a} = 2.8 \times 10^{-9} \ (\text{s Pa})^{-1}, \quad \gamma_{pc,c} = 1.0 \times 10^{-9} \ (\text{s Pa})^{-1}, \quad \gamma_{pv,v} = 6.0 \times 10^{-11} \ (\text{s Pa})^{-1}.$$

For the exchanges between compartments corresponding to the branching of blood vessels, we use

$$\gamma_{c,a} = \frac{B_{\text{flow}}}{\Delta p_{c,a}|\Omega|} = 3.3 \times 10^{-6} \ (\text{s Pa})^{-1}, \quad \gamma_{v,c} = \frac{B_{\text{flow}}}{\Delta p_{v,c}|\Omega|} = 9.7 \times 10^{-6} \ (\text{s Pa})^{-1}.$$

where $B_{\text{flow}}$ = 116mL/100g/min (from [58]), and $\Delta p_{c,a}$, $\Delta p_{v,c}$ correspond to the blood pressure drops between vessels. We assume a $\Delta p_{c,a}$ = 40mmHg blood pressure drop from arteries to capillaries and a $\Delta p_{v,c}$ = 13mmHg blood pressure drop from capillaries to veins.

Similarly, to obtain the transfer coefficients for connected PVS compartments, we assume that the CSF flow in PVS is proportional to CSF production $Q_{\text{CSF}}$ = 3.38 $\mu$L/min. This latter value from [59] corresponds to an upper estimate of the CSF production in our case since more recent works using another technique of measurement found 1.40 $\mu$L/min for CSF production rate [60]. Using the production rate of CSF as flow in the PVS is, of course, an upper estimate of what the actual flow in the network is. Then, assuming $\Delta p_{pa,pc}$ = 1mmHg and $\Delta p_{pc,pv}$ = 0.25mmHg, we arrive to

$$\gamma_{pa,pc} = 1.83 \times 10^{-7} \ (\text{Pa s})^{-1}, \quad \gamma_{pc,pv} = 7.31 \times 10^{-7} \ (\text{Pa s})^{-1}.$$

To compute the exchange coefficients between the pial surface artery PVSs and the arterial PVS as well as for the exchange between ECS and SAS, we adapt the fluid resistance coefficient for this space from the one used in [48] to obtain (see S1 Appendix)

$$L_{\text{PVSpial},pa} = 1.25 \times 10^{-6} \ (\text{s Pa})^{-1}, \ \text{and} \ L_{e,\text{SAS}} = 3.13 \times 10^{-7} \ (\text{s Pa})^{-1}.$$

The osmotic pressure in the capillary compartment has been reported to be 20mmHg [61]. We thus set $\pi_a = \pi_c = \pi_v$ = 20mmHg and $\pi_e = \pi_{pa} = \pi_{pc} = \pi_{pv} = 0.2 \times \pi_c$ (extravascular osmotic pressures have been chosen from the fact that due to the BBB the osmotic pressure in the ECS within the brain is known to be lower than 30% of the capillary one [61]).

We now define the advective mass exchange coefficients using the equation

$$\tilde{\gamma}_{j,i}^{^{14}\text{C}-\text{inulin}} = \gamma_{j,i}\left(1 - \sigma_{ij,\text{reflect}}^{^{14}\text{C}-\text{inulin}}\right),$$

where $\sigma_{ij,\text{reflect}}^{^{14}\text{C}-\text{inulin}}$ is the reflection coefficient for the $^{14}$C-inulin and the membrane under consideration. Since Inulin is approximately 5000Da in size (measured in [62]), we set

$$\sigma_{ij,\text{reflect}}^{^{14}\text{C}-\text{inulin}} = 0.2,$$

for all the membranes.

The diffusive permeabilities through the astrocyte endfeet (AEF) membrane for $^{14}$C-inulin test case 2 and 3 are computed from [63, 64] (see S1 Appendix for details)

$$P_{pa,e}^{14\text{C–inulin}} = P_{pv,e}^{14\text{C–inulin}} = 1.2 \times 10^{-3} \text{ mm s}^{-1}, \quad P_{pc,e}^{14\text{C–inulin}} = 4.1 \times 10^{-4} \text{ mm s}^{-1}.$$

From the two previous parameters, we defined for the 7 compartments system $\gamma_{pa,a}$, $\gamma_{pc,c}$, $\gamma_{pv,v}$, $\gamma_{e,pa}$, $\gamma_{e,pc}$, $\gamma_{e,pv}$ and $\gamma_{e,v}$ as well as their corresponding transfer coefficients for $^{14}$C-inulin.

The transfer of solutes between vessel compartments for which the connection exists without a membrane is assumed to be solely driven by convection, and the fact that $^{14}$C-inulin does not cross the BBB implies

$$P_{a,pa}^{14\text{C–inulin}} = P_{v,pv}^{14\text{C–inulin}} = P_{c,pc}^{14\text{C–inulin}} = P_{a,c}^{14\text{C–inulin}} = P_{c,v}^{14\text{C–inulin}} = P_{pa,pc}^{14\text{C–inulin}} = P_{pc,pv}^{14\text{C–inulin}} = 0,$$

For connected vessel compartments, the solvent-drag reflection coefficient is assumed to be $\sigma_{\text{reflect}} = 1$.

Altogether, we obtain the transfer coefficients reported in Table 2.

The last value we specify is the CSF volume surrounding the brain, *i.e.* in the subarachnoid space. This parameter value is required to define the boundary conditions. The reported values for this volume vary in the literature, ranging from $90\mu L$ [66] to $520\mu L$ [67], but seem to be consistently in the region 5–20% of the total intracranial volume. For the simulations in this paper, we will assume that the CSF volume is 10.8% of the total intracranial volume, as reported by [68]. Assuming that the intracranial volume consists of brain tissue and the CSF spaces, this value corresponds to a CSF volume of $V_{\text{CSF}} = 0.12 \times |\Omega|$, where $|\Omega|$ is the volume of the brain tissue.

**Remark 3.** *In the previous section, all the parameter values required to model the clearance of $^{14}$C-inulin using Eq (1) or System (3) have been specified. Coefficients for which no value has been specified are assumed to be zero, e.g. for exchange coefficients between compartments that are not in communication.*

**Remark 4** *Most parameter values have been found using measurements from in-vitro or in-vivo biological experiments. However, we have indicated the ones for which the values are adapted from the literature or the works from which we extracted the values and estimated these parameters using numerical simulations. We recall that S1 Appendix provides details about the estimates and computed coefficients.*

### 2.4 Model variations

**Sensitivity analysis.** Some of the parameters used in this model may be uncertain or, to some degree, unknown. It is not clear a-priori how these uncertainties may affect the output from

**Table 2. Baseline diffusive and convective exchange parameters.**

| Symbol | Unit | Meaning | Value | Reference |
|---|---|---|---|---|
| $\gamma_{i \to j}$ | 1/(Pa s) | Fluid mass transfer coefficient | $\gamma_{a,e} = 2, 7 \times 10^{-9}, \gamma_{v,e} = 6 \times 10^{-11}, \gamma_{c,e} = 9 \times 10^{-10}$ | Computed |
| | | | $\gamma_{pa,e} = 2.19 \times 10^{-7}, \gamma_{pv,e} = 1.95 \times 10^{-7}, \gamma_{pc,e} = 9.20 \times 10^{-9}$ | |
| | | | $\gamma_{a,pa} = 2.76 \times 10^{-9}, \gamma_{v,pv} = 6.00 \times 10^{-11}, \gamma_{c,pc} = 9.98 \times 10^{-10}$ | |
| | | | $\gamma_{a,c} = 3.14 \times 10^{-6}, \gamma_{c,v} = 9.65 \times 10^{-6}$ | |
| | | | $\gamma_{pa,pc} = 1.83 \times 10^{-7}, \gamma_{pc,pv} = 7.31 \times 10^{-7}$ | |
| | | | $\gamma_{\text{PVSpial},pa} = 1.25 \times 10^{-6}, \gamma_{e,SAS} = 3.13 \times 10^{-7}$ | |
| $\tilde{\gamma}_{i \to j}^{14\text{C–inulin}}$ | 1/(Pa s) | Advective mass transfer coefficient | Given by Eq (22) | |
| $\lambda_{i \to j}$ | s$^{-1}$ | Solute mass transfer coefficient | $\lambda_{pa,e}^{14\text{C–inulin}} = 3.70 \times 10^{-3}, \lambda_{pv,e}^{14\text{C–inulin}} = 3.72 \times 10^{-3}, \lambda_{pc,e}^{14\text{C–inulin}} = 3.70 \times 10^{-3}$ | Computed from [65] |

**Table 3. Factors of variation for each of the tested parameters.**

| Parameter | Factors of variation |
|---|---|
| $\kappa_e$ | {0.1, 1, 10, 100, 1000} |
| $\kappa_{pa}$ | {0.01, 0.1, 1, 10, 100, 1000} |
| $\kappa_{pc}$ | {0.01, 0.1, 1, 10, 100, 1000} |
| $p_{\mathrm{SAS}}$ | {0.5, 0.75, 1, 1.5, 2} |
| $P_{pa,e}^{14\mathrm{C-inulin}}$ | {0.01, 0.1, 1, 10, 100, 1000} |
| $\gamma_{pa,a}$ | {0.1, 0.5, 1, 2, 5} |
| $D^*$ | {0.5, 0.75, 1, 1.5, 2} |
| $\phi_{pa}$ | {0.1, 0.5, 1, 10, 20} |

the model. We therefore set out to investigate the model sensitivity to changes in unknown or critical parameters. Each parameter was varied, taking into account the relative uncertainty found in the literature. In particular, we changed ECS permeability ($\kappa_e$) by a factor between 0.1 and 1000, periarterial permeability ($\kappa_{pa}$) by a factor 0.01 to 1000, pericapillary permeability ($\kappa_{pc}$) by a factor 0.01 to 1000, CSF pressure ($p_{\mathrm{SAS}}$) by a factor 0.5 to 2, diffusive transfer between the periarterial and extracellular network ($P_{pa,e}^{14\mathrm{C-inulin}}$) by a factor 0.01 to 1000. In addition, we tested a change in apparent diffusion $D^*$ by a factor of 0.5 to 2 and a change in periarterial porosity between 0.1 and 50.

For the 7-compartment model, we further changed the BBB permeability between the arterial and periarterial compartments ($\gamma_{pa,a}$) by a factor of 0.1 to 5. For a complete overview of all parameter variations for the sensitivity analysis, see Table 3. To save computational cost, the sensitivity analysis was performed on the mesh of resolution 16, which yield a few % faster clearance than the 32 mesh used in the simulations (for details see S3 Appendix).

**The effect of CSF clearance.** It has been suggested that the flow of CSF in the SAS plays a major role in clearance also from the brain parenchyma [69, 70]. In the present study, the effect of CSF clearance from the SAS is modelled by three different boundary conditions for the concentration: 1) A homogeneous Dirichlet condition as described by Equation (2.2), representing instantaneous clearance from the SAS, 2) CSF/ISF exchange and conservation of $^{14}$C-inulin in the intracranial compartment (Eq (13)), and 3) CSF/ISF exchange and exponential decay of particles from the SAS due to CSF production and absorption (Eq (12)).

**The effect of sleep.** Xie *et al.* [33] reported an increase of the ECS porosity when the animal is sleeping, which may increase convective transport in the brain [13]. Indeed, they indicated that the porosity of the ECS in the awake state is $\phi_e^{\mathrm{awake}} = 0.14$ whereas, in the sleeping state, they measured $\phi_e^{\mathrm{sleep}} = 0.23$. Using the Kozeny-Carman equation, this leads to the relation (see [71] for example)

$$\kappa_e^{\mathrm{sleep}} = 5.5 \times \kappa_e^{\mathrm{awake}}.$$

Recent results [72] indicate that when the animal is asleep, dilation and reduction of the perivascular spaces are observed due to vasomotion. Assuming that the vasomotion leads on average to an enhancement of PVS porosities and that the contraction of the blood vessels leads to a constant factor $C_\phi$ of increase of porosity for these spaces *i.e.*

$$\phi_j^{\mathrm{sleep}} = C_\phi \phi_j^{\mathrm{awake}}.$$

Then, assuming free (Poiseuille) fluid flow in perivascular spaces, a change of porosity creates a modification of the permeability leading to

$$\kappa_j^{\text{sleep}} = C_\phi^2 \kappa_j^{\text{awake}},$$

(see S1 Appendix for details). We assume that the parameter values corresponding to the awake state are given by the baseline values of Table 1. Based on the measurements from [72], we use an upper estimate of PVS variations during sleep and assume $C_\phi = 4$.

**The effect of communication with blood.** The blood vessels composing the cerebral vasculature are not completely impermeable, and there is a debate going on to which extent CSF/ISF communicates with the microcirculation [38]. We consider both a 4-compartment model (test case 2) that assumes no communication between CSF/ISF and blood, and a 7-compartment model (test case 3) where fluid can exchange between the blood vessels and the perivascular spaces around them.

## 2.5 Quantities of interest

To study how sample size variations from experimental data could affect the results, we integrate the concentrations over several cubes $\omega \subset \Omega$ of varying sizes embedded in the brain mesh to represent possible measurement samples of the brain. In addition, we assess the mass of $^{14}$C-inulin in the entire brain $\Omega$.

For the first test case (ECS only), the relative mass of $^{14}$C-inulin in the entire brain at time $t$ is denoted by

$$c_{tot}(t) := \frac{\int_\Omega \phi_e c_e(t, \mathbf{x}) \, d\mathbf{x}}{\int_\Omega \phi_e c_e(0, \mathbf{x}) \, d\mathbf{x}}.$$

The relative mass of $^{14}$C-inulin in the ECS within a cube $\omega \subset \Omega$ centred around the injection point is defined as

$$c_\omega(t) := \frac{\int_\omega \phi_e c_e(t, \mathbf{x}) \, d\mathbf{x}}{\int_\omega \phi_e c_e(0, \mathbf{x}) \, d\mathbf{x}}.$$

For the other test cases (4-, and 7-compartments), the relative mass of $^{14}$C-inulin in the entire brain and within a cube $\omega \subset \Omega$ at time $t$ is denoted by

$$c_{tot}(t) := \frac{\int_\Omega \sum_{j \in J} \phi_j c_j(t, \mathbf{x}) \, d\mathbf{x}}{\int_\Omega \phi_e c_e(0, \mathbf{x}) \, d\mathbf{x}}, \quad c_\omega(t) := \frac{\int_\omega \sum_{j \in J} \phi_j c_j(t, \mathbf{x}) \, d\mathbf{x}}{\int_\omega \phi_e c_e(0, \mathbf{x}) \, d\mathbf{x}},$$

respectively.

We further measure the fluid velocity in the different compartments. From the solution of the pressure equations, we compute the vector fields

$$\mathbf{u}_j = -\frac{\kappa_j}{\phi_j \mu_j} \nabla p_j, \quad j \in J, \tag{15}$$

to obtain the velocity inside the $j$-th compartment. From these computed velocity fields $\mathbf{u}_j$, we compute the average velocity within a compartment $u_{\text{aver},j}$ and the maximal velocity $u_{\text{max},j}$

given by

$$u_{\text{aver},j} = \frac{\int_\Omega \sqrt{\mathbf{u}_j \cdot \mathbf{u}_j}\, d\mathbf{x}}{|\mathbf{\Omega}|} \quad u_{\text{max},j} = \|\sqrt{\mathbf{u}_j \cdot \mathbf{u}_j}\|_{L^\infty}, \tag{16}$$

To compute the volume of fluid transferring between compartment $j$ and compartment $i$, we use

$$Q_{j,i} = \int_\Omega \gamma_{j,i}(p_i - p_j)\, d\mathbf{x}. \tag{17}$$

To compute the volume of CSF exchanged between compartment $j$ and the SAS, we use

$$Q_{j,\text{SAS}} = \int_{\partial\Omega} \left( -\frac{\kappa_j}{\mu_j} \nabla p_j \cdot \boldsymbol{\nu} \right) ds. \tag{18}$$

To compute the mass of $^{14}$C-inulin moving from compartment $i$ to $j$, we use [37]:

$$M_{ji}(t) = \int_\Omega \lambda_{j,i}(c_i - c_j) + \frac{(c_j + c_i)}{2}\tilde{\gamma}_{j,i}(p_i - p_j - \sigma_{i,j}(\pi_i - \pi_j))\, d\mathbf{x}. \tag{19}$$

## 2.6 Computational mesh, solution method and verification

The computational mesh used for the simulations in this paper was constructed from the "Waxholm Space Atlas of the Sprague Dawley Rat Brain v4" (RRID: SCR_017124) [73–75], available under the licence CC-BY-SA 4.0 (https://creativecommons.org/licenses/by-sa/4.0/) at https://www.nitrc.org/projects/whs-sd-atlas. The atlas provides a detailed segmentation of different regions within the rat brain.

In the original study behind the atlas [73], the animal was anaesthetized by intraperitoneal injection of a mixture of Nembutal (Ovation Pharmaceuticals, Inc., Lake Forest, IL) and butorphanol, and transcardially perfused with 0.9% saline and ProHance (10:1 v:v) for 4 minutes followed by a flush of ProHance in 10% phosphate-buffered formalin (1:10 v:v). All procedures and experiments in their work were approved by the Duke University Institutional Animal Care and Use Committee [73].

Since the models in this paper do not separate between tissue from different regions of the brain, the segmentation is mainly of interest for removing unwanted sections. Most importantly, we wanted to remove the segments representing various parts of the ventricles. Moreover, we removed some external artefacts such as the spinal trigeminal tract, the optic nerves, and parts of the auditory system [74].

The various segments in the raw data file have a few irregularities. For example, in regions where the lateral ventricles are very thin, small groups of unlabeled voxels create holes in the 3D reconstruction of the ventricles. To repair these irregularities, we have made use of **3D Slicer** (https://www.slicer.org/), an open-source software application for visualization and analysis of medical images [76]. 3D Slicer provides a segment editor with tools for manual labelling of voxels, hole filling and surface smoothing. After refining the segmentation of the ventricular system, it may be removed from the original volume to create a realistic representation of the brain surface. The surface is exported as an stl-file to be used in the meshing algorithm.

The creation of the computational mesh is performed by SVMTK (https://github.com/SVMTK/SVMTK), which provides a python API for 3D mesh generation methods from the CGAL library. The mesh generation algorithm consists of a Delaunay refinement process

followed by an optimization phase [77]. Following the procedures described in [78], we created the mesh illustrated in Fig 2a.

To solve the Eqs (1) and (3), we use the finite element method for the discretization in space and an implicit Euler method to integrate the resulting ordinary differential systems in time.

In this paper, we choose a resolution for the spatial mesh of $h = 1/32$. The temporal domain is $[0, T]$ with $T = 360$min with a time step of $\Delta t = 1$min. Details of the mesh and time resolutions can be found in Appendix C.2 in S3 Appendix. The numerical scheme has been implemented using the FEniCS Library [79, 80], and the linear system was solved using the generalized minimal residual method (GMRES) and the incomplete LU (ILU) preconditioner. Our code is publicly available on GitHub at the following link: https://github.com/jorgenriseth/multicompartment-solute-transport.

## 3 Results

### 3.1 CSF flow in the 4-compartment model

Fig 3 depicts the pressure fields inside the different compartments for the 4-compartment model. We observe that for baseline parameter values, the pressure gradients in the different fields give a bulk flow of fluid in line with the glymphatic theory. Indeed, using Eq (15), our model represents an inflow of CSF from the surface of the brain in the PVS of arteries and an outflow from the PVS of veins. Smaller pressure gradients leading to lower velocities directed from the surface to the depth of the brain are also seen in the ECS and the PVS of capillaries.

Computing the transfer of CSF between the compartments using Eq (17), we obtain

$$Q_{pa,e} = 0.72\,\mu\text{L/min}, \quad Q_{e,pv} = 0.27\,\mu\text{L/min}, \quad Q_{e,pc} = 4.4 \times 10^{-3}\,\mu\text{L/min}.$$

The transfer between the compartments and the SAS is computed in the same way using Eq (18), and we obtain

$$Q_{\text{SAS},e} = 0.22\,\mu\text{L/min}, \quad Q_{\text{SAS},pa} = 0.94\,\mu\text{L/min}, \quad Q_{pv,\text{SAS}} = 0.68\,\mu\text{L/min}.$$

In this notation, we choose subscripts such that the flow occurs from the first denoted compartment to the second (e.g. flow occurs from the PVS of arteries to the ECS).

From these pressure fields, we compute the velocity of the CSF in the compartments using Eq (15). We report the average velocities $u_{\text{aver}}$ and the maximal ones $u_{\text{max}}$ for each compartment in Table 4.

### 3.2 Transport within the brain

In the following two subsections, we report the relative amount of $^{14}$C-inulin in the entire brain from the diffusion and the 4-compartment simulations using Eqs (10) with (12) as standard boundary conditions (aside from Subsection 3.2.4 the boundary condition used for the concentration equations will always be (10) with (12) and will be referred to as "Decay" boundary conditions). We then vary the size of the measurement sample (*i.e.* the domain in which the remaining mass of $^{14}$C-inulin is computed) and the boundary conditions.

**3.2.1 Diffusion in the ECS only.**   Pure diffusion steadily decreased the tracer amount found within the brain over the entire simulation time, and ∼53% of the tracer remains after 6 hours (Fig 4a, blue dashed line). If we assume an exponential decay between the first- and the last time point, the clearance corresponds to a rate constant of 0.0018/min. Fig 4c shows the distribution of $^{14}$C-inulin transported by pure diffusion (*i.e.* Eq (1)) in the ECS at different points in time. The tracer spreads radially out from the point of injection, and peak

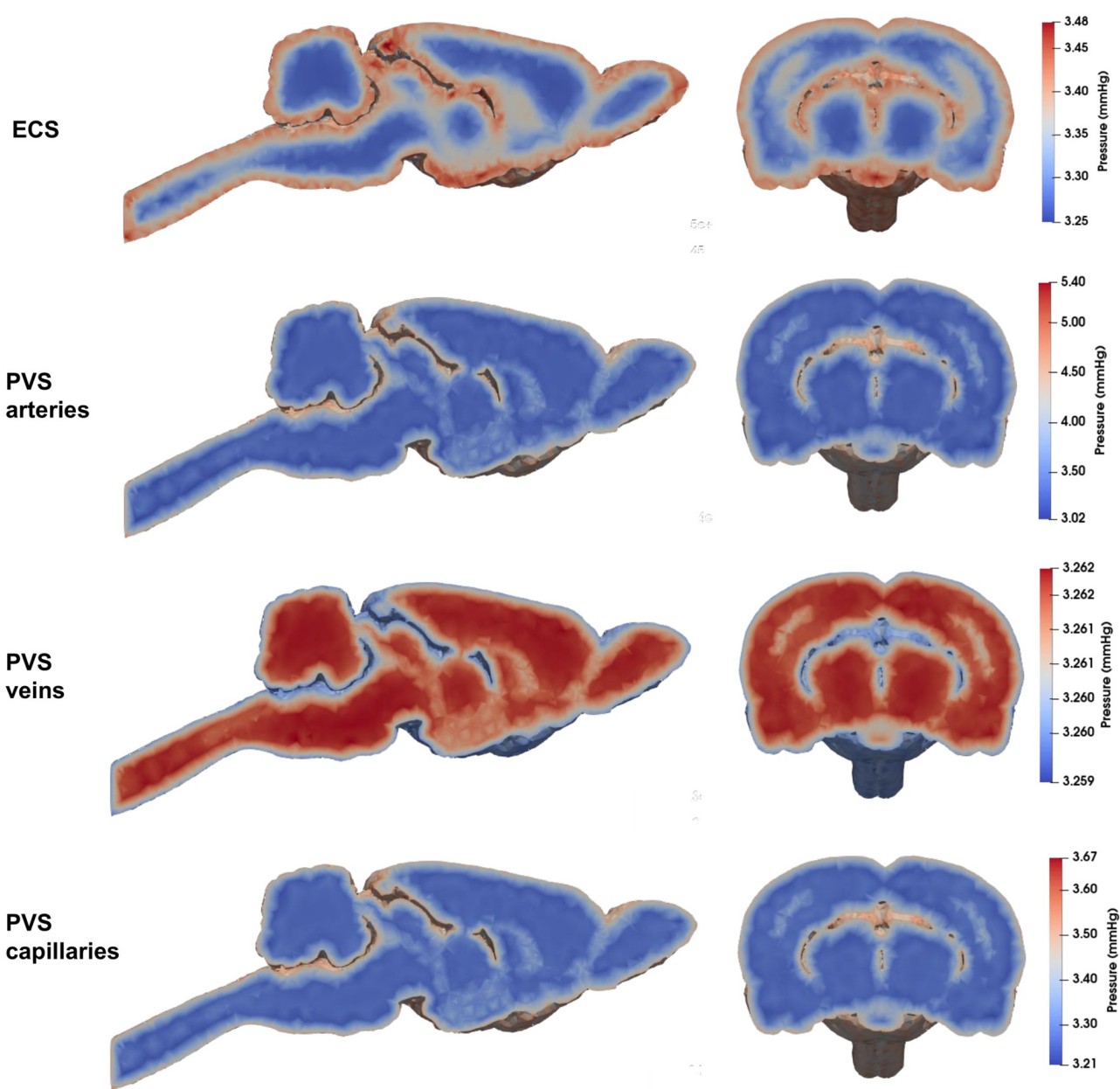

**Fig 3. Pressure fields in the 4 compartments (left: coronal cut, right: sagittal cut).**

**Table 4. Velocities of CSF in the different compartments for baseline parameter values.**

| Compartment | $u_{\text{aver}}$ (in $\mu$m/s) | $u_{\text{max}}$ (in $\mu$m/s) |
|---|---|---|
| PVS arteries | $9.5 \times 10^{-1}$ | 7.9 |
| ECS | $3.3 \times 10^{-3}$ | $6.0 \times 10^{-2}$ |
| PVS veins | $4.4 \times 10^{-1}$ | 3.0 |
| PVS capillaries | $4.0 \times 10^{-3}$ | $2.7 \times 10^{-2}$ |

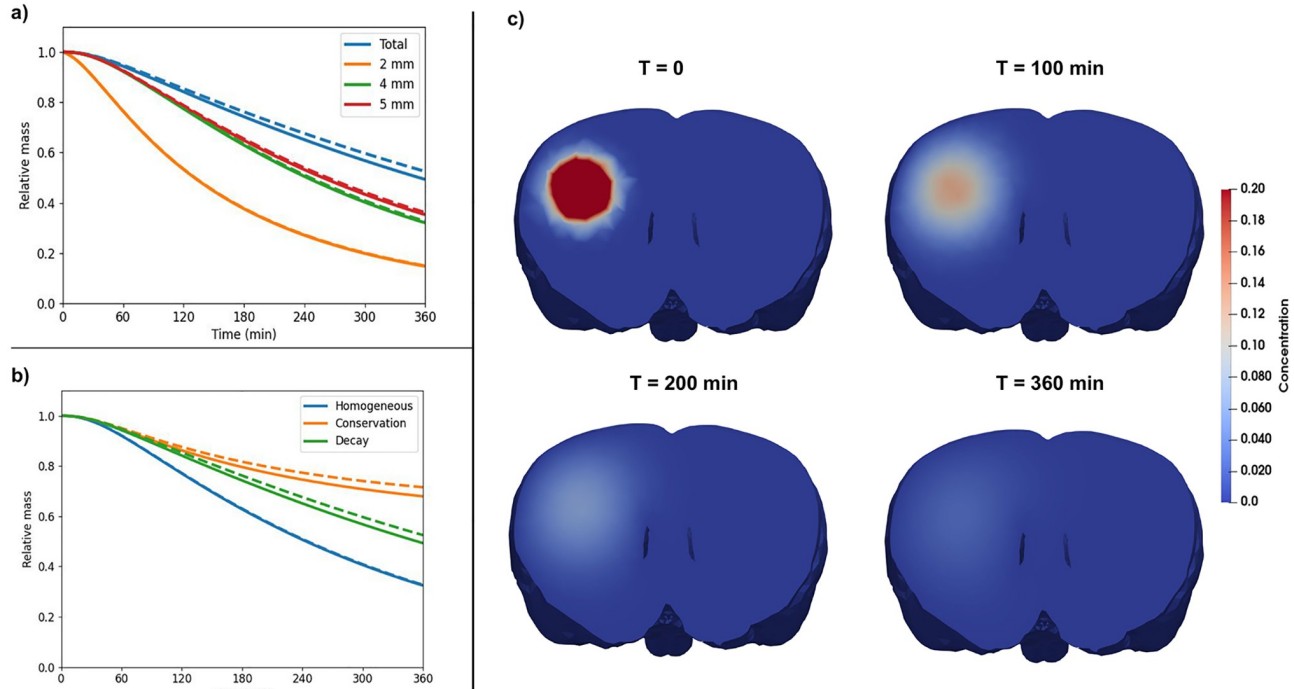

**Fig 4.** a) Relative [14]C-inulin mass located within regions of varying size surrounding the injection point. Solid lines result from the multi-compartment model simulations, while dashed lines result from diffusion only in the ECS. b) Relative [14]C-inulin mass located in the totality of the brain for the different boundary conditions. Solid lines result from the multi-compartment model simulations, while dashed lines result from diffusion only in the ECS. c) Evolution in space and time of [14]C-inulin relative concentration in the ECS for test case 1 (single diffusion). The colour scale is chosen for a visual comparison between all time points.

concentration has decreased drastically after T = 360 min. At the first time step, some very small negative values appear near the tail of the Gaussian curve, but are smoothed out over time.

**3.2.2 4-compartment convection-diffusion.** Fig 5 shows the spatial distribution of [14]C-inulin concentration over time in all 4 compartments considered in [14]C-inulin test case 2. Initially, the tracer is contained only in the ECS where it was first injected. Already after 10 minutes, the concentration spreads equally to all compartments. From all time points on, the tracer spreads radially outwards in all compartments, similar to the test case for pure diffusion. We note here that even with equal concentrations, the total mass of tracer differs between each compartment due to differences in porosity. [14]C-inulin is thus mainly still contained to the ECS in the 4-compartment model. The tracer in the 4-compartment convection-diffusion model is cleared from the brain slightly faster compared to diffusion alone and ~ 50% of the tracer remains in the brain after 6 hours, corresponding to a rate constant of 0.0023/min.

**3.2.3 Effect of the measurement sample.** Fig 4a shows the evolution of the relative mass of [14]C-inulin inside the rat brain and in samples of the brain of different sizes (cubes of side length 2 mm, 4 mm, and 5 mm). The boundary conditions for the concentration equations correspond to the time-dependent Dirichlet boundary conditions (12). For the smallest measurement sample, the relative mass of tracers remaining in the sample after 6 hours were ~15% for diffusion and for the 4-compartment model (compared to 53% and 50% for the entire brain). We observe that as the measurement sample size increases, the mass of [14]C-inulin remaining in the sample increases.

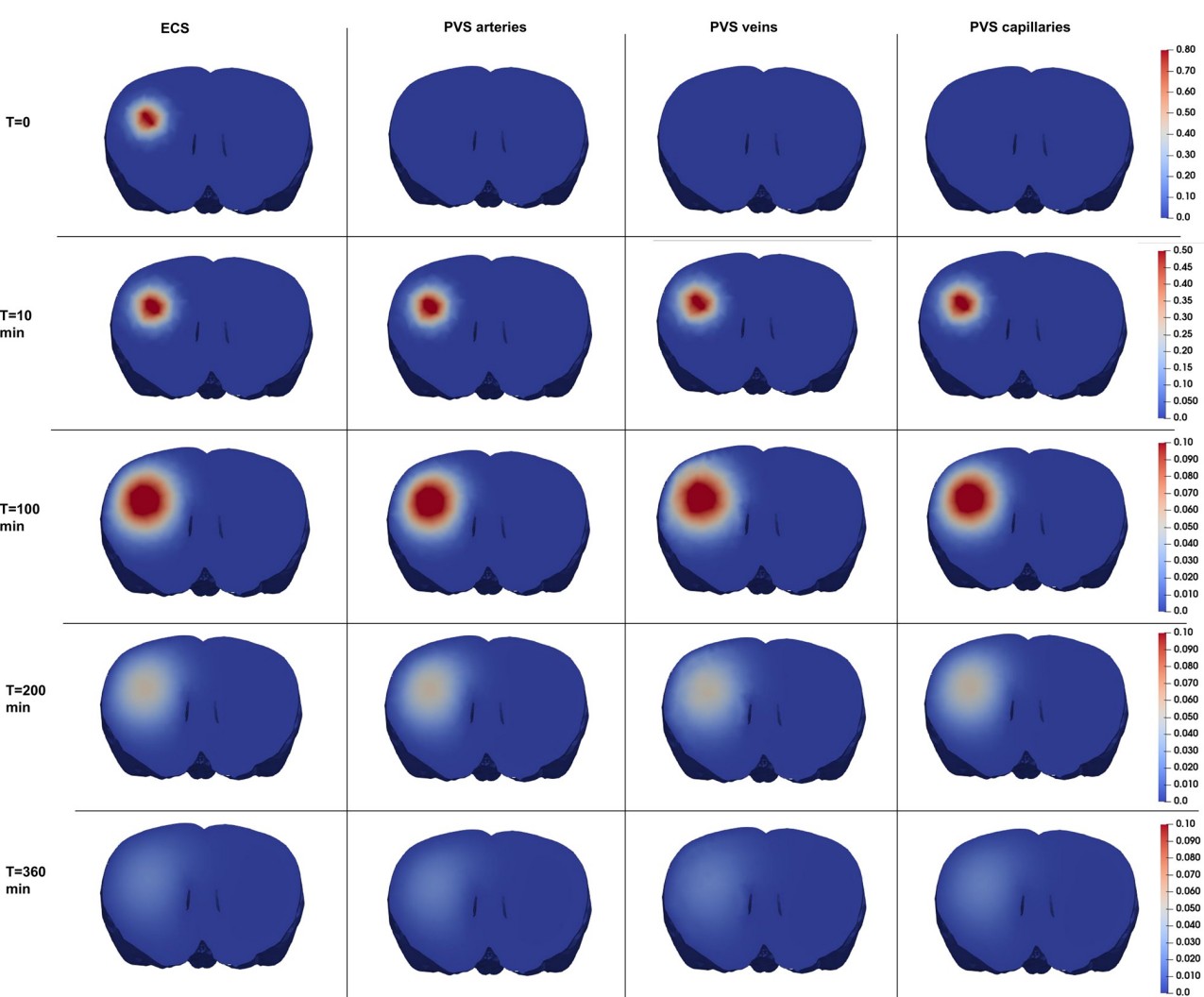

**Fig 5. Evolution in time and space of the relative $^{14}$C-inulin amount in the rat brain (frontal cut at the injection point) within the 4 compartments of test case 2.**

**3.2.4 Effect of the concentration in the subarachnoid space.** Fig 4b shows the evolution of the relative mass of $^{14}$C-inulin for the three different boundary conditions for the concentration equations: Homogeneous Dirichlet boundary condition, conservation of the mass in the subarachnoid space (corresponding to Eqs (10) with (14)), and clearance of $^{14}$C-inulin in the subarachnoid space (corresponding to Eqs (10) with (12)). Fig 4b compares the relative mass of tracer for the diffusion model (dashed lines) to the four-compartment model (solid lines). In both models, the homogeneous Dirichlet boundary conditions lead to fast clearance from the tissue with ∼33% remaining in the brain after 6 hours (For both diffusion only and the 4-compartment simulations). When the concentration of $^{14}$C-inulin is computed using the time-dependent Dirichlet boundary conditions representing tracer conservation in the SAS, the mass of tracers is close to plateau level at 68% or 72% at 6 hours (corresponding to a clearance rate of ∼0.001/min). With the time-dependent boundary conditions modelling absorption of CSF in SAS, the relative tracer mass steadily decreases and ends up between the two previously described cases with 50–53% of the tracer remaining in the brain after 6 hours.

### 3.3 Variations of the 4-compartment model

**3.3.1 Sensitivity analysis.** For several parameters, value changes of several orders of magnitude do not drastically alter the results. A full overview of model sensitivity changes in parameters is shown in Table 3. In particular, for the extracellular and pericapillary permeabilities ($\kappa_e$, $\kappa_{pc}$), subarachnoid space pressure ($p_{SAS}$)) and the periarterial to extracellular diffusive transfer ($P_{pa,e}$), there is less than 1% difference in tracer mass between the tested parameter values.

An increase of 1000 in periarterial permeability reduces the tracer mass by 10%, while a decrease gives no difference in output. With a change in the diffusion constant, the final tracer mass increases by 43.2% when the diffusion coefficient is decreased by a factor 2 and decreases by 36.2% when the diffusion coefficient is increased by a factor 2. An increase in periarterial porosity slightly delayed clearance, and with an increase of a factor 20, the tracer mass at the final timestep is increased by 7.4%.

Figs 8 and 9 in S2 Appendix show results from a systematic parameter variation for both the 4- and the 7-compartment model.

**3.3.2 Effect of an increase in ECS porosity.** It is postulated that sleep has an effect on the clearance of solutes in the brain [33]. The accepted hypothesis is that the ECS porosity increases during sleep, enhancing the convection in this space and even dominating diffusion [13, 33]. This is better measured by the Péclet number $Pe$ that measures the importance of convection over diffusion ($Pe < 1$ if diffusion dominates while $Pe > 1$ if convection is preponderant).

With an increase of ECS porosity from 0.14 to 0.23, we find no relevant difference in the total CSF transfer between the compartments. Interestingly, we find that the maximum velocity in the ECS increases to $u_{\max} = 7.9 \times 10^{-2} \mu$m/s (from $6.0 \times 10^{-2} \mu$m/s) and the average velocity of CSF in ECS increases to $4.0 \times 10^{-3} \mu$m/s (from $3.3 \times 10^{-3} \mu$m/s). See Table 4 for all reference velocities computed with baseline parameter values. The Péclet number in ECS increases from $3.2 \times 10^{-2}$ for baseline coefficients to $3.9 \times 10^{-2}$ after ECS porosity increase.

Tracer clearance is slightly slower for the four-compartment model when ECS porosity is increased (blue versus orange line, Fig 6). Since the velocity field in the ECS is directed inwards from the brain surface, solutes are transported away from the sinks at the domain boundaries. Hence, additional flow in the ECS slows down clearance in this compartment, and the relative mass of tracers within the brain is, in this case, 57% after 6 hours.

**3.3.3 Effect of an increase in PVS porosity.** Increasing the PVS porosity by a factor of 4 decreases the clearance slightly from the brain via PVS. The relative mass of tracers found in the brain after 6 hours increases from 50% during baseline to 53% with increased PVS porosity (Fig 6, blue versus red line). Since the diffusive transfer between the compartments tends to average the concentration between them, increasing the porosity of PVSs increases the mass of $^{14}$C-inulin in these compartments. Since the PVS of arteries is now larger and is an inflow route (with a convective field directed to the depth of the brain), clearance of $^{14}$C-inulin appears slower.

**3.3.4 Combined enhancement of the extracellular volume and perivascular spaces.** Combining the increase of both ECS and PVS porosity and permeability, we obtain the following computed amount of CSF transfer between the compartments

$$Q_{pa,e} = 1.88\,\mu\text{L/min}, \quad Q_{e,pv} = 9.5 \times 10^{-1}\,\mu\text{L/min}, \quad Q_{e,pc} = 2.0 \times 10^{-2}\,\mu\text{L/min},$$

$$Q_{e,\text{SAS}} = 7.7 \times 10^{-1}\,\mu\text{L/min}, \quad Q_{\text{SAS},pa} = 3.5\,\mu\text{L/min}, \quad Q_{pv,\text{SAS}} = 2.3\,\mu\text{L/min}.$$

We also obtain the maximum and averaged velocities reported in Table 5.

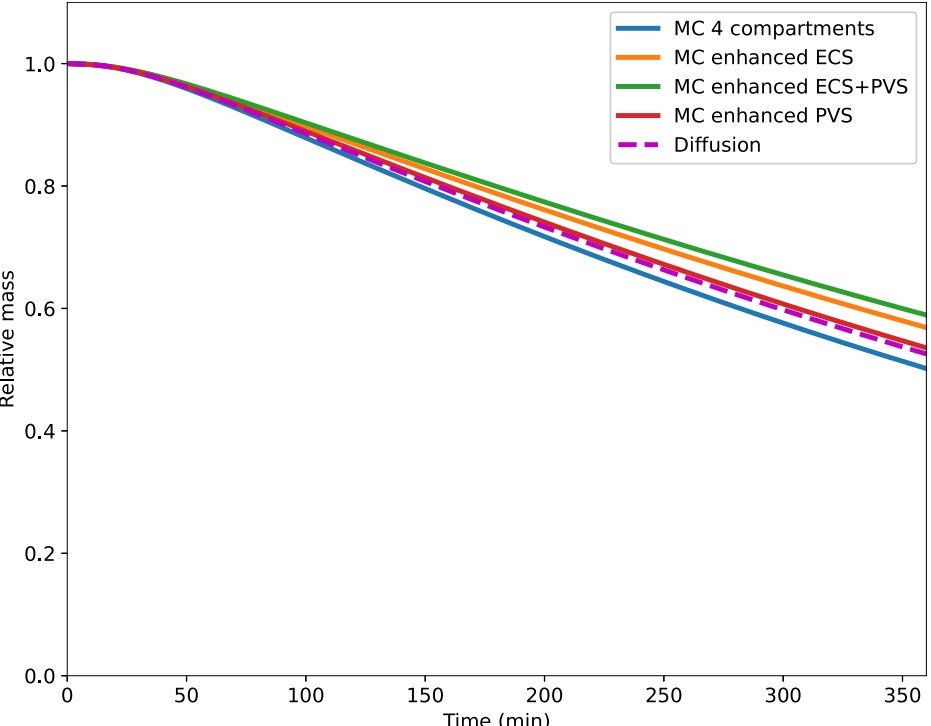

**Fig 6. Comparison of $^{14}$C-inulin clearance for different variations of porosity and permeability coefficients.** "MC Baseline" denotes the clearance curve given by the multi-compartment model with baseline parameter values and is hidden by the dashed curve "Diffusion", representing the clearance given by the application of the Diffusion model in the ECS compartment only. The enhancement of ECS porosity leads to the curve denoted "MC enhancement ECS" and the increase of the porosities in all the compartments gives the clearance curve denoted "MC enhancement ECS +PVS"

With an increase in both ECS and PVS permeability, we observe a very similar clearance compared to when the ECS porosity is increased (Fig 6, orange versus green line). The mass of $^{14}$C-inulin within the brain after 6 hours for the 4-compartment model, with increased porosity in ECS and PVS, reaches ∼58% of the original content.

### 3.4 7-compartment model: Additional effect of cerebral blood perfusion

Using the baseline parameter values for the second and the third test cases, we obtain the pressure fields in the ECS shown in Fig 7a. Interestingly, the leakage of fluid from arteries and capillaries to the PVSs occurring in the 7-compartment model changes the pressure fields compared to the 4-compartment model (shown in Fig 3), in which the PVSs were assumed to be isolated from the blood. In contrast to the 4-compartment model, the fluid flow in the PVS

**Table 5. Velocities of CSF in the different compartments for an increase of porosity and permeability in all the 4 compartments.**

| Compartment | $u_{\text{aver}}$ (in $\mu$m/s) | $u_{\text{max}}$ (in $\mu$m/s) |
|---|---|---|
| PVS arteries | 1.8 | 9.2 |
| ECS | $9.1 \times 10^{-3}$ | 0.12 |
| PVS veins | 0.73 | 4.6 |
| PVS capillaries | $7.0 \times 10^{-3}$ | $2.0 \times 10^{-2}$ |

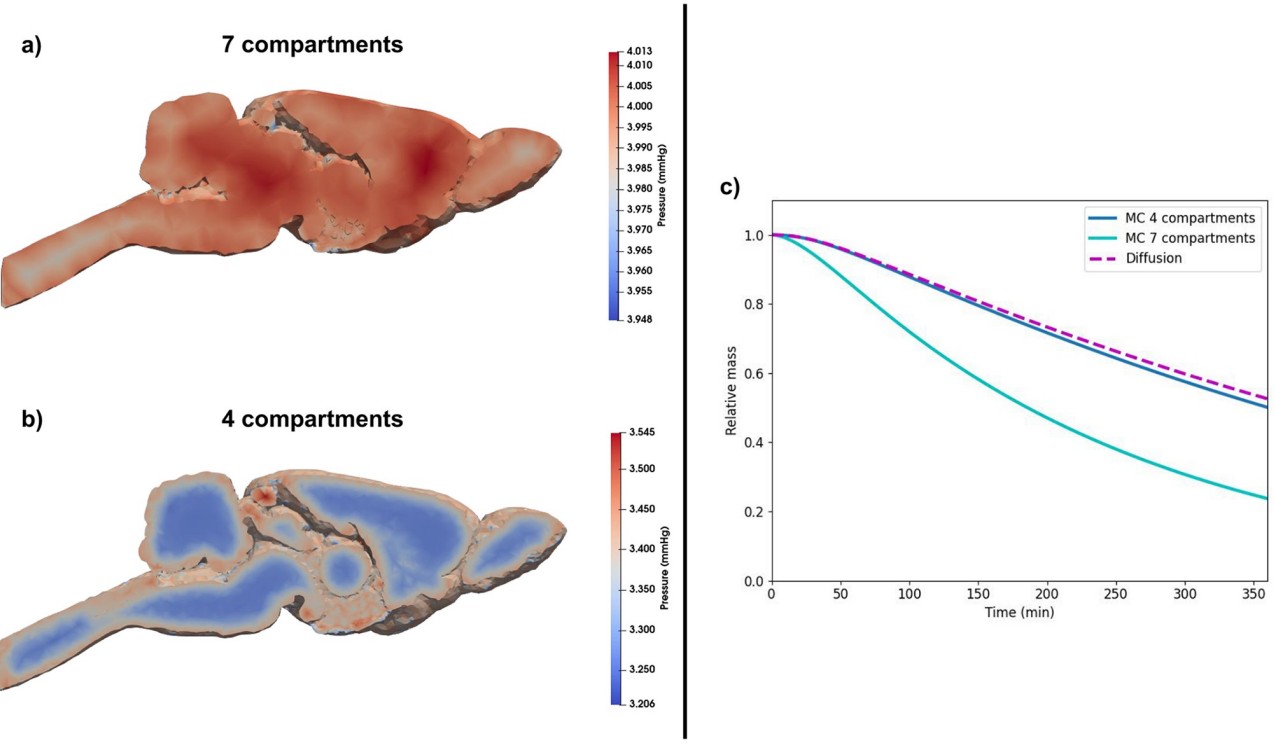

**Fig 7.** a-b) Comparison of pressure in ECS for the test cases 3 a) and 2 b). The velocity field is directed opposite of the gradient of pressure. Thus the velocity is mostly oriented to the outside of the brain, and the magnitude is larger when blood is considered in the model. c) Comparison of $^{14}$C-inulin clearance for test case 2 with baseline parameter values and increase of ECS and PVSs porosities with test case 3. "MC Baseline" denotes the clearance curve given by the multi-compartment model with baseline parameter values. The enhancements of ECS and PVSs porosities lead to the curve denoted "MC enhancement ECS+PVS", and the result of test case 3 is denoted "MC 7-compartments"

of arteries and the ECS is directed towards the brain surface. In addition, flow velocities are increased compared to the 4-compartment model (see Table 6 for details).

Fig 7b shows the clearance curves of $^{14}$C-inulin obtained with all three test cases (pure diffusion, 4-compartment, 7-compartment). We observe that with the additional effect of blood perfusion, the clearance is much faster compared to both pure diffusion and all variations of the 4-compartment model. Only ~23% of the tracer remains in the brain after 6 hours for the 7-compartment model (compared to 53% and 50% for pure diffusion and the 4-compartment model). The clearance rate for the 7-compartment model with baseline parameter values is thus 0.0041/min, which is close to twice the clearance rate for the 4-compartment model, see Subsection 3.2.2).

**Table 6. Velocities of CSF and blood in the different compartments for baseline values coefficients for test case 3.**

| Compartment | $u_{\text{aver}}$ (in $\mu$m/s) | $u_{\text{max}}$ (in $\mu$m/s) |
|---|---|---|
| Arterial blood | $3.88 \times 10^3$ | $69 \times 10^3$ |
| Venous blood | $88 \times 10^1$ | $5.6 \times 10^3$ |
| Capillary blood | 1.2 | 28 |
| PVS arteries | 0.69 | 5.8 |
| ECS | $4.3 \times 10^{-3}$ | $7.1 \times 10^{-2}$ |
| PVS veins | 2.7 | 18 |
| PVS capillaries | $2.8 \times 10^{-3}$ | $1.9 \times 10^{-2}$ |

Computing the fluid flow between the different compartments using Eq (17), we find

$$Q_{a,pa} = 2.6\,\mu L/min, \quad Q_{v,pv} = 4.2 \times 10^{-3}\,\mu L/min, \quad Q_{c,pc} = 2.3 \times 10^{-1}\,\mu L/min,$$

$$Q_{pa,e} = 1.29\,\mu L/min, \quad Q_{e,pv} = 7.5 \times 10^{-1}\,\mu L/min, \quad Q_{e,pc} = 1.9 \times 10^{-2}\,\mu L/min,$$

$$Q_{a,\text{influx}} = 2.3\,mL/min, \quad Q_{v,\text{outflow}} = 1.8\,mL/min, \quad Q_e, \text{SAS} = 3.5\,\mu L/min,$$

$$Q_{\text{SAS},pa} = 6.9 \times 10^{-1}\,\mu L/min, \quad Q_{pv,\text{SAS}} = 2.0\,\mu L/min.$$

**3.4.1 Sensitivity analysis.** Similar to the 4-compartment model, we found that results were robust to changes in several model parameters. For all parameter changes except changes in the arterial to periarterial fluid transfer ($\gamma_{pa,a}$), the diffusion coefficient ($D$) and the periarterial porosity ($\phi_{pa}$), the tracer mass in the brain at the final time step changed less than 3.1% compared to baseline parameters. As the total mass is already low at the final time step due to rapid convective transport, a constriction in fluid transport from the blood vessels to the periarterial space had a drastic effect. Changing this transport parameter by a factor of 0.1 increased the mass at the final timestep by 82.3%. An increase in the transport parameter by a factor of 5 had an opposite but still drastic effect, with a 96.8% reduction in tracer mass compared to the baseline case. Increasing the diffusion coefficient by a factor of 2 led to a 19.7% decrease in tracer mass after 6 hours, while decreasing the diffusion by a factor of 2 led to an increase of 21.1%. Changes in periarterial porosity had a smaller effect with a 2.0% decrease in tracer mass when $\phi_{pa}$ was reduced by a factor of 0.1, and an increase of 29.4% when the porosity was increased by a factor of 20.

## 4 Discussion

The main goal of this article is to propose a multi-compartment model representing fluid movement and solute transport in the brain. We apply our model to the glymphatic system at the scale of the rat brain. We design our model and numerical method to explore different scenarios and hypotheses related to the clearance of $^{14}$C-inulin from the brain. Indeed, changing the parameter values for permeability, porosity, and exchange coefficients allows us to represent, for example, the possible effect of sleep, the disruption of a membrane, an enhanced CSF flow in the parenchyma or the effect of blood perfusion on the standard picture depicted by the glymphatic theory. Furthermore, the numerical results explore different situations and allow us to assess the importance of different modelling aspects, such as the boundary conditions, and experimental biological aspects, such as the importance of the measurement sample. To the best of the authors' knowledge, this is the first attempt at using a multi-compartment model to combine fluid flow and transport of solute at the scale of the entire brain. This work is largely built upon works related to blood perfusion in tissues [23–25].

### 4.1 Effect of the measurement sample

The effect of the measurement sample is depicted in Fig 4a. Our results show that if the measurement sample is small, clearance appears to be faster compared to larger samples or the entire brain. This information needs to be taken into account when quantitatively comparing biological experiments to simulations (e.g. clearance curves). For instance, comparisons between simulation results and the results obtained by, e.g. Iliff et al. [1] and measurements in a piece of tissue or slice (e.g. tracer influx in Xie et al. [33]) is not straightforward. According to simulations of transport in mice [14], diffusion is quantitatively comparable to experimental

data on Aqp4 null mice with relative A$\beta$-concentration. However, with a convective velocity field, simulations match the wild-type mice experiments. We argue that our results can not be compared to the results from Ray *et al.* [14] due to the difference in measurement sample. When measuring the entire brain, clearance curves differ from those measured in a cube of arbitrary size, as shown in Fig 4a).

## 4.2 Modelling the clearance of $^{14}$C-inulin from the SAS using boundary conditions

Usually, mathematical models representing clearance of $^{14}$C-inulin from the brain use homogeneous Dirichlet boundary conditions for the concentrations (*e.g.* see [2, 16]). This modelling assumes that clearance in the SAS is instantaneous, which in reality, is not the case. Some studies have taken this into account by adding mass conservation between the brain and the SAS [17]. However, the numerical results presented in Fig 4b show that taking the concentration of solutes in the CSF in the SAS into account leads to much slower clearance rates (there are 39% less relative $^{14}$C-inulin mass cleared assuming conservation of $^{14}$C-inulin in SAS than for homogeneous Dirichlet boundary conditions after 6 hours). Even when adding an absorption rate of CSF in SAS, we also obtain slower clearance rates compared to homogeneous Dirichlet boundary conditions (There is a difference of 20% of relative $^{14}$C-inulin mass after 6 hours between the boundary conditions modelling slower clearance from the CSF in the SAS and the homogeneous boundary condition). Hence, our results indicate that forthcoming mathematical models should be careful with the choice of boundary conditions to obtain biologically relevant results.

## 4.3 Baseline parameter values for the 4-compartment model

Even though the model includes many parameters, most of them can be estimated using measurements reported in the literature. Using baseline parameter values, diffusion in the ECS (test case 1) gives clearance results very similar to the ones given by the 4-compartment model considering the PVSs as being isolated from the effect of blood perfusion (test case 2). The results from these models correspond qualitatively to the clearance results reported in Xie *et al.* [33], but the absolute clearance is slightly slower. Measuring both absolute recovery and rate constant, Xie *et al.* report an absolute recovery of $\sim 60\%$ and a rate constant of around 0.006. Regarding recovery, we have similar values, but for the rate constant we find a value of only 0.002/min. This difference may be explained by the lack of full recovery (*i.e.* a steady state plateu level) in the experimental data. Overall, these results indicate that diffusion in the ECS is the main mechanism to explain the observed clearance, but also that an additional mechanism is needed. Most of the $^{14}$C-inulin mass is contained and cleared within the ECS. Therefore, even though the Péclet number is higher in the PVSs than in the ECS (Pe = 9.4 in the PVS of arteries compared to Pe = $1.6 \times 10^{-2}$ in the ECS), most of the transport still occurs in the latter. We also note that the maximal velocity in PVS arteries obtained from baseline coefficient values (see Table 4) is close to the measurements of CSF velocity in the PVS arteries at the pial surface in [6, 7] ( 7.9$\mu$m/s in our work compared to 18$\mu$m/s in Mestre *et al.* [7]). We also note that the location of the maximal velocity we obtained is close to the surface of the brain where the gradient of the pressure is the largest in the PVS arteries compartment (see Fig 3). ISF velocity has also been reported to be around 0.1 − 0.25 $\mu$m/s by Cserr *et al.* [81, 82] and numerical results obtained in [83] indicated a peak velocity in the ECS close to the surface of the brain and with magnitude $u_{\max} = 0.5\mu$m/s. Similar fluid velocities in the parenchyma have been reported by Nicholson [84] and Abbott [85] while a numerical study [17] have shown that even a velocity of 1$\mu$m/s may play a complementary role in transport within the

parenchyma. Ray *et al.* [14] obtained using a computational model a bulk flow velocity in the ECS of $0.008 - 0.42\mu$m/s. Altogether, our model compares well with the previously cited results with an average velocity in the ECS of $u_{\text{aver}} = 0.0033\mu$m/s and a peak velocity located close to the surface of the brain of $0.06\mu$m/s (see Table 4). We note that our average velocities are slightly lower than experimental results. However, movement in the parenchyma is typically measured by lumping all compartments together while we report each compartment separately. This difference may explain the discrepancy between our simulations and experimental data of flow in the ECS.

## 4.4 Increasing the porosity of the ECS slows the clearance of $^{14}$C-inulin

Since the work of Xie *et al.* [33], sleep is believed to play an important role in the clearance of $^{14}$C-inulin. In [33], an increase in the porosity of the ECS was measured when the animal was asleep. Our results show that when only the ECS porosity was increased, the clearance of $^{14}$C-inulin was slower. This may be explained by the fact that increasing the ECS porosity leads to smaller concentration gradients, hence decreasing diffusive movement. However, in this scenario, we assumed that the diffusion coefficient remained constant. Furthermore, the increased ECS porosity only led to a 74% increase in average velocity, and still, the Péclet number remains small (Pe = $2.2 \times 10^{-2}$ after ECS porosity increase) in the ECS.

Compared to the usual representation of the glymphatic system in which vessels are clearly spaced, and convective movement occurs between them, our multi-compartment model represents this effect through the exchange terms. The enhancement of the ECS porosity allows for more fluid transfer from the ECS to PVS veins (as shown in Subsection 3.3.4), hence, capturing well the hypothesized faster convective movement in the ECS from the PVS of arteries to the PVS of veins during sleep. However, the usual schematic representation of the glymphatic system does not allow us to consider the directions of the pressure gradients in the ECS at the scale of the brain. If we assume that the convective movement in the PVS of arteries is generated by a pressure gradient, our results show fluid flow in the ECS directed inwards from the surface of the brain. This latter counters the diffusive movement and, hence, slows down the clearance of solutes. This could indicate an effect that is neglected in the current glymphatic theory and needs further investigation.

The minor effect of increased porosity in our model seems to indicate that to obtain the measured effect of sleep (see Xie *et al.* [33]), another induced change must take place. Recent results from [72] indicate that sleep also induces vasomotion in the brain. If we assume that sleep induces a general vasoconstriction trend, we can model this effect by reducing the radius of blood vessels, and hence, the PVS width increases. Therefore, porosity and permeability coefficients are adapted correspondingly (see S1 Appendix). This leads to an enhanced CSF movement in all structures (e.g. an increase of 305% of fluid volume from the PVS of arteries to the ECS) and affects the clearance of $^{14}$C-inulin. The clearance curves shown in Fig 6 clearly reveal that the clearance is slower for the scenario in which PVS permeabilities are increased with a magnitude associated with sleep. When the permeability is further increased, our sensitivity analysis revealed a nonlinear effect, and faster clearance is observed. This indicates that the vasomotion of arteries only could contribute positively to clearance, and that a multi-compartment system such as the brain may involve complex, non intuitive interaction between compartments. However, it is worth mentioning that even when varying key parameters such as ECS, periarterial and pericapillary permeability with orders of magnitude we did not obtained rate constants as observed by Xie *et al.* [33] for $^{14}$C-inulin in sleeping animals (rate constant of $\sim$0.015/min). These results indicate that the improvement of clearance due to

sleep does not seem to be explained only by an increase of the porosity coefficients in the ECS and PVSs.

## 4.5 Fluid leakage from the blood vessels improve $^{14}$C-inulin clearance and make the periarterial space an outflow route

Using biologically relevant parameter values, our results indicate that if the effect of leakage from the blood vessels is taken into account, the flow of CSF in the ECS and PVS of arteries is reversed compared to the standard picture of the glymphatic theory. With the inclusion of blood vessels, the flow direction is in line with the proposed hypothesis by Cserr et al. [86]. The PVS arteries compartment becomes an outflow route in this case. Additionally, the flow is also reversed in the ECS compared to the 4-compartment model (as observed in Fig 7a and 7b). This leads to a faster clearance of $^{14}$C-inulin as observed in Fig 7c. The fluid velocities in all compartments are all increased in this case compared to the 4-compartment model case. We obtain a Péclet number of $4.1 \times 10^{-2}$, which still indicates that diffusion in the interstitial space is the dominant effect for transport. However, referring to Croci *et al.* [17], we emphasize that even a small increase in velocities in the ECS could lead to a significant change in transport that could explain our enhanced clearance for that case. The sensitivity analysis also revealed that increasing the diffusion coefficient has less effect on the 7-compartment model, suggesting that convection play more prominent role in this case. Indeed, for this third test case, the relative amount of $^{14}$C-inulin decays exponentially with $\sim$20% of relative $^{14}$C-inulin mass after 6 hours. The shape for the clearance curve corresponds more to the sleeping animals' results from Xie *et al.* [33], however with standard parameters, the rate constant in our model was around 3–4 times lower than the rate constant observed experimentally. However, the sensitivity analysis revealed that increased blood filtration (by a factor around 100) in the 7-compartment model gave a rate constant of $\sim$0.014/min. Therefore, combining our results from the 4- and 7-compartment models seem to indicate that the transfer of fluid between blood vessels and PVSs and ECS provide a great potential to increase clearance during sleep. This could be related again to observed vasomotion of cerebral vessels during sleep [72].

### 4.6 Limitations and further works

Our model is based on a homogenization procedure that represents the different structures (ECS, PVS, blood vessels) as a continuum. Usually, the derivation of the macroscopic model is assumed to be correct if the ratio between the length scale of the pores (in our case the distance between vessels) and the length scale of the chamber is less than one. In our model, this holds true for most compartments (see Shipley *et al.* [24] for homogenization related issues). However, we emphasize that this ratio is close to one for cerebral arterioles and venules. This strong modelling assumption could be relaxed by considering a 1D-3D model (see *e.g.* [87]) in which only the ECS and capillaries are represented as continuous media while the other structures are modelled by one-dimensional curves within the domain.

1D-3D models provide a more detailed description of the pressure-field and solute transport. However, this type of model requires very high resolution meshes with cell sizes on the scale of the radius of the blood vessels to properly model fluid and solute exchange between tissue and the 1D structures [88]. It is therefore currently too computationally costly to be used at the scale of the whole brain. Nevertheless, some of the observations and assumptions from this present work could be tested and verified more accurately with such a 1D-3D model. The comparison between these two types of models will be the subject of a future work.

In our article, the clearance of solutes from the CSF in the SAS is taken into account using a simplified boundary condition. Indeed, we assumed that once the solute reaches the SAS it

diffuses instantaneously within the CSF in this region. We plan to derive more rigorously these boundary conditions in a future work to first model fluid movement inside a three dimensional subarachnoid space and then to seek effective boundary conditions while studying the asymptotic limit of zero width for the subarachnoid space.

Furthermore, due to the complexity induced by the modelling of the different compartments and exchange between them, there are 8 coefficients per compartment (some of them might be shared between two compartments, for example for the exchange through a shared membrane). Some measurements of these parameters exist, however sometimes in different species (rats versus humans), and we have to the best of our ability translated parameters to reflect rat physiology. In addition, the measured values may suffer from experimental uncertainties. The present model could also be used to investigate phenomena with continuous production and clearance of substances that are naturally produced in the brain [89, 90]. However, for this application additional parameters would be needed, and given the already complex parameter space, this addition were therefore not considered in the present study.

To study uncertainties about parameter values, we performed a sensitivity analysis. However, inverse modelling may also be used to find parameter values numerically. Optimization techniques have been previously used to estimate parameters in the context of the glymphatic system [11]. To optimize parameters, PDE constrained optimization techniques (see *e.g.* [91] about inverse problems and [92] for PDE-constrained optimization) have to be applied to our model to minimize the error between the output of our model and results from experiments. Two main difficulties arise in our case. First, the dimension of the inverse problem (the number of parameters to optimize) is very large but based on our sensitivity analysis and on experimental works, some of them can be fixed and, hence, decrease the computational cost of the optimization by reducing the dimension. The second issue comes with experimental data and whether sufficient data is available to perform parameter optimization on our model.

In our study, we made some variations of parameter values to model a possible increase in ECS or PVSs volumes. The enhancement of ECS volume is reported in [33]. However, the increase of PVSs volume has been measured recently in [72] and does not appear to be fixed but rather a time-dependent value. Indeed, oscillations, vasoconstrictions and vasodilatations may occur over a few minutes (see Fig 2d in [72]). If the effect of these oscillations in porosity were accounted for in our model, the equations would change drastically ($\phi(t)$ becomes a time-dependent function and stays in the time derivative of both the concentration and pressure equations). Adding the effect in the system forces us to keep the time derivative in the pressure equation and solve a coupled system of equations at each time step for each compartment. This will increase the computational cost of the simulations tremendously.

The use of standard continuous finite elements for the discretization of the diffusion equation leads to the presence of small oscillations of the numerical solution. See e.g. [78] for details about this effect and some remarks about stabilization, which could be included in further works. However, we note that integrated quantities over large domains (e.g. the brain) are not affected as small oscillations around zero concentration even out. In this work, we arranged the scale on the figures so that no negative values for the concentration appear visually.

## 5 Conclusion

In this paper, we presented a multi-compartment model for fluid and solute transport with application to the glymphatic system of a rat brain. The model allows us to test the effect of different physiological changes (e.g. sleep) and assess different theories concerning fluid flow and transport in the brain. Unless blood filtration was added to the model, diffusion was the main

driving force for transport. However, as our simulations show, only a small leakage from blood vessels increased clearance by an order of magnitude.

## Supporting information

**S1 Appendix. Computing biologically relevant parameters** [2, 29, 32, 48–50, 58–60, 63, 64, 93–101].
(PDF)

**S2 Appendix. Sensitivity analysis.**
(PDF)

**S3 Appendix. Numerical verification** [102, 103].
(PDF)

## Author Contributions

**Conceptualization:** Alexandre Poulain, Jørgen Riseth, Vegard Vinje.

**Data curation:** Alexandre Poulain, Jørgen Riseth.

**Formal analysis:** Alexandre Poulain, Jørgen Riseth.

**Methodology:** Vegard Vinje.

**Project administration:** Vegard Vinje.

**Software:** Alexandre Poulain, Jørgen Riseth.

**Supervision:** Vegard Vinje.

**Validation:** Jørgen Riseth.

**Visualization:** Alexandre Poulain, Jørgen Riseth.

**Writing – original draft:** Alexandre Poulain, Vegard Vinje.

**Writing – review & editing:** Alexandre Poulain, Jørgen Riseth, Vegard Vinje.

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
