## [Decision Letter · Decision Letter 0]

31 Oct 2022

PONE-D-22-27008Multi-compartmental model of glymphatic clearance of solutes in brain tissuePLOS ONE

Dear Dr. Vinje,

Thank you for submitting your manuscript to PLOS ONE. After careful consideration, we feel that it has merit but does not fully meet PLOS ONE’s publication criteria as it currently stands. Therefore, we invite you to submit a revised version of the manuscript that addresses the points raised during the review process.It would be useful to point out the more important uncertainties and to carry out a sensitivity analysis.It is better to discuss about using your modelling in estimating physiological glymphatic parameters from dynamic contrast MRI measurements.The authors have not validated the model and it is better to perform pomparison using experimental measurements.Please submit your revised manuscript by Dec 15 2022 11:59PM. If you will need more time than this to complete your revisions, please reply to this message or contact the journal office at plosone@plos.org. Please include the following items when submitting your revised manuscript:A rebuttal letter that responds to each point raised by the academic editor and reviewer(s). You should upload this letter as a separate file labeled 'Response to Reviewers'.A marked-up copy of your manuscript that highlights changes made to the original version. You should upload this as a separate file labeled 'Revised Manuscript with Track Changes'.An unmarked version of your revised paper without tracked changes. You should upload this as a separate file labeled 'Manuscript'.

We look forward to receiving your revised manuscript.

Kind regards,

Quan Jiang, Ph,D.

Academic Editor

PLOS ONE

Journal Requirements:

4. Please upload a new copy of Figure 6 as the detail is not clear. Please follow the link for more information:

https://blogs.plos.org/plos/2019/06/looking-good-tips-for-creating-your-plos-figures-graphics/

https://blogs.plos.org/plos/2019/06/looking-good-tips-for-creating-your-plos-figures-graphics/

Reviewers' comments:

Reviewer's Responses to Questions

**Comments to the Author**

1. Is the manuscript technically sound, and do the data support the conclusions?

Reviewer #1: Yes

Reviewer #2: Yes

Reviewer #3: Yes

Reviewer #4: No

2. Has the statistical analysis been performed appropriately and rigorously? 

Reviewer #1: N/A

Reviewer #2: N/A

Reviewer #3: N/A

Reviewer #4: N/A

3. Have the authors made all data underlying the findings in their manuscript fully available?

Reviewer #1: Yes

Reviewer #2: Yes

Reviewer #3: Yes

Reviewer #4: Yes

4. Is the manuscript presented in an intelligible fashion and written in standard English?

Reviewer #1: Yes

Reviewer #2: Yes

Reviewer #3: Yes

Reviewer #4: Yes

5. Review Comments to the Author

Reviewer #1: This interesting paper presents a discretized, “compartment” model of the so-called glymphatic system for brain clearance, consisting of advection and diffusion occurring in CSF and ISF in various parts of the rat brain. This is an ambitious effort to address an important feature of brain physiology and health. The model seems to be constructed with considerable care, and the results are of considerable interest. I recommend that the paper be published, but I have several questions and suggestions for the authors to consider.

General comments:

The model is applied to the clearance of a finite amount of tracer injected at an initial time, treated as an initial value problem in which the total amount of tracer in the brain decreases with time. On the other hand, the glymphatic system operates naturally to clear metabolic waste molecules that are produced continually in the brain, and this production and clearance are in balance and produce a steady state. Can the authors relate their model to such a steady-state clearance of metabolic waste?

The authors have carefully chosen numerical values of all of the parameters in the model, based on the best experimental data. However, some of these parameter values are subject to considerable uncertainly. It would be useful to point out the more important uncertainties. It would be quite useful to carry out a sensitivity analysis on the model, identifying the parameters for which the model is most sensitive to uncertainty in their values. An example of such a sensitivity analysis is that of Boster et al. 2022 (J. Roy Soc. Interface 19: 20220257), who performed a sensitivity analysis of the hydraulic network model of the glymphatic system described in Tithof et al. 2022 (your reference 72).

Specific comments:

Abstract: Don’t capitalize glymphatic or biology.

Page 2, par. 1: Reference 60 also argues that transport in the ECS is not by diffusion alone. Further theoretical evidence in favor of a flow of ISF in the ECS, based on the sleep/wake variation in clearance, is given in Thomas 2022, FBCNS 19:30.

Page 2, first two paragraphs: Fluid dynamic models, and in particular, previous compartment models, of CSF flow and transport are discussed in a very recent review article, Kelley and Thomas 2023, Annual Review of Fluid Dynamics, pre-publication version available online.

Page 3, Remark 1: The correct term here is “homogeneous”, not “isotropic”. Isotropic means independent of direction, and you have already assumed isotropy by assume scalar, rather than tensor, permeability, etc.

Page 7, section 2.3.2, Porosity coefficients: For the mouse brain, although the porosity of PVSs around penetrating arteries and arterioles is unknown, the PVSs around pial arteries have been shown to be essentially open spaces (Min Rivas et al. 2020, J. Roy. Soc. Interface, 17: 20200593), i.e., porosity = 1. Assuming this is also true in the rat brain, can you account for this in your model?

Page 10, Remark 3: “precised”? Do you mean “specified”?

Page 10, The effect of sleep: The effect of sleep/wake variations in permeability is discussed in Thomas 2022, cited above.

Page 13, Table 3 and text: Here you might want to compare with the flow speeds in pial PVSs in the mouse brain, measured by particle tracking in Mestre et al. 2018 (your reference 47). They are comparable to your values.

Page 15, section 3.3.1: Here, and perhaps elsewhere, you might want to point out the relative importance of advection and diffusion in the ECS in your various scenarios, best expressed in terms of the Peclet number.

Reviewer #2: This works develops a mathematical multi compartment framework for modeling solute clearance from the brain by the convective and diffusion flow through the perivascular and extracellular, (and vascular) spaces. The paper is well written and straightforward to follow. The authors did an excellent job to include many physiological parameters into the modelling. However, I have two recommendations to improve the paper.

1) The results of this kind of modelling are very dependent on the prior knowledge of the model’s parameters. As appreciated by the authors, the experimental driven physiological parameters used in this model (Tables 1 and 2) are not precise and can vary considerably from one experiment to another. Moreover, even changing the structural parameters of the modeling (like the mesh size, number of compartments, etc.) can change the results [As studied in the results section]. Therefore, I would recommend doing a sensitivity analysis (although the authors mentioned this in their future plans, page 19) to study the sensitivity of the results to the input parameter variations. This may help researchers to know which parameter has a more dominant effect on the results.

2) I would like the authors to discuss (in one or two paragraphs) about using this modelling in estimating physiological glymphatic parameters from dynamic contrast MRI measurements, like previous works such as (Lee et al.,J. Neurosci, 2015) and (Davoodi-Bojd et al., Neuroimage, 2019). Do the authors think their modelling can be used in a backward manner?

Minor comments:

1- The authors should provide a rational why they chose 14C-insulin for their modelling.

2- Page 4, “where |Ω| =R Ω 1 dx = 2313 mm3 is the brain volume”. Please put a reference for the value of the brain volume.

3- Page 5, please explain why the tracer distribution in the initial condition is assumed Gaussian? This implies there is no spatial restriction.

4- Page 5, Boundary condition. It is known that the blood flow (and hence pressure) is changed during heart cycle, which cause vascular pulsation. Although it is hard to include this parameter in the modelling, but I would like the authors to elaborate on this parameter.

5- Page 6, “A decrease of molecules within the brain, corresponds to an increase of concentration in the SAS, and vice-versa.”. First, I assume “molecules” means the solute molecules (14C-Insulin), right? Secondly, I don’t understand term “vice-versa” here. Does this mean solute molecules return back to the brain?

6- In the text there are several places the authors mentioned “molecules”. Please clarify if it refers to 14C-insulin.

Reviewer #3: Overall this is a well-produced and well-written manuscript that presents a new model of inter- and intra-compartmental solute transport and flows within and between the CSF and interstitial fluid of the brain, including effects of both diffusion and convection, as well as the porosity of these spaces. It could provide a very good tool for studying and evaluating aspects of the proposed glymphatics system and their feasibility, with one of the most interesting findings being how big an effect fluid leakage from larger blood vessels could have on the clearance of solutes (here 14C-inulin) from the brain. The manuscript is also clear with respect to data availability (e.g., code and meshes) and ethical considerations (where applicable, since it is a model study). Appropriate mesh and time step analyses have been performed. I do have some minor comments that I think would improve the manuscript.

Minor comments:

1) It is a bit unclear from figure 1 how flows are limited. In the figure description it says that fluid movement occurs along the red arrows, but couldn’t the arrows from the PVS to the ECS be bidirectional? Especially for the final simulation, where PVS flow directions change?

2) I also think that, in the beginning, it is a bit unclear that the SAS is handled through boundary conditions, as it does not seem to be included explicitly as a “compartment”. This should be clarified or emphasized more, early on. Is it possible to complement figure 1 with a more complete view of the entire model?

3) Some single data values lack references (or the references are unclear). I am thinking of the p_pv, p_pvspial pressures for example, otherwise state that (if and how) they are calculated. I am also wondering about the ICP and ISF pressure references, since they are far apart in time and in different groups of rats. How representative are they? Are there no simultaneous measurements of ISF pressure and “ICP” in rats published? Here a sensitivity analysis would be appropriate to mention. Sensitivity analyses are mentioned in the limitations. Could they not be included in the current manuscript without problem?

4) There are some small typos and grammatical errors here and there that can be adjusted (the manuscript lacks some final polish).

5) Figure 6 is missing its labels

6) The description of Figure 1 includes AEF and BBB but no such labels are seen in the figure.

7) Write explicitly that you are using the Laplace operator in the RHS of eq. (1) (diffusion equation. Most readers will know what it is but some may read it as a delta (gradient and divergence cannot be mistaken however).

8) Similarly, I think that e.g. the Darcy’s law should be referenced before equation 2 to add a pedagogical extra step to improve readability making it easier for any reader to assimilate the equations. At the very least a bit more info on where the equations come from would be an improvement.

9) On page 8 the pressure drop from arteries to capillaries (and from capillaries to veins) are not the same as on page 22 (and the indexes are shifted, and the Pc-v is named Pa-c on page 22).

10) On page 15, for the simulation where ECS porosity is increased, the velocity field seems directed towards the ventricles, as the authors state. Do the authors find this reasonable/physiological? It is mentioned but not really discussed in the discussion section (and that increases in the porosity of ECS seem to act in the opposite direction to increases in PVS. Reasonable?). The results are very useful regardless, however some additional comments would be appropriate.

11) Also, I may have missed it, but are the calculated flow velocities in the PVS compared or validated against other estimations in the literature? (The comparisons of clearance to that in the literature are good, however)

12) I have a hard time understanding the first paragraph of the limitation section where the authors bring up the 1D-3D approach. Could the authors clarify the main differences of this model compared to the current one and especially how it is more computationally costly?

Reviewer #4: This manuscript presents numerical simulations of glymphatic clearance of brain solutes, that is, clearance by the combined effects of diffusion and advection by flowing cerebrospinal fluid. A novel numerical model is described, in which the authors treated the brain as a set of interpenetrating, communicating fluid compartments: extracellular space, periarterial space, perivenous space, pericapillary space, arteries, veins, and capillaries. In the model, all compartments co-exist throughout the brain without spatial separation, and all are treated as porous media, such that effects of small-scale channel geometries are estimated by bulk porosity and permeability coefficients. Transport of fluid and solutes among compartments is presumed to occur through membranes. In some simulations, blood compartments are not explicitly considered. The authors present pressure fields and spatiotemporal solute evolution as predicted by the simulations. From the results, the authors conclude that clearance rates depend on the size of the sample considered, that solute boundary conditions in the subarachnoid space surrounding the brain affect predicted clearance rates, that the chosen parameter values seem to correspond to transport during wakefulness as observed by Xie et al. (Sleep drives metabolite clearance from the adult brain, Science, 2013), that increasing the porosity (volume) of extracellular space slows clearance, and that allowing fluid outflow from blood compartments causes fast clearance because fluid is driven to flow toward the brain surface in the extracellular and perivascular spaces.

This numerical model is quite different from others used previously to study glymphatic clearance and thus has potential to bring new insights. Unfortunately, the authors have not validated the model. Though grid convergence studies and time step size effects were considered for a few cases, those necessary steps are not sufficient to show that a model as complicated as this one is valid and trustworthy. I have less concern about the numerical methods than about the dozens of biophysical values which must be chosen as input parameters: porosities, permeabilities, pressures, and more. Few have been measured directly. The chosen values often depend on complicated reasoning involving many simplifying assumptions (see [Supplementary-material pone.0280501.s001]). Though that reasoning usually seems credible, the authors must go further by demonstrating that the values and the model accurately reproduce real, established biology. The model must be used to solve a problem whose answer is well-known. The problem should involve multiple compartments, porous media, and solute transport by both diffusion and advection. Experimental measurements should be used for comparison; solutions from prior models might also be used. Simulating cases where closed-form analytic solutions are available would also help, though they are likely to be drastically simpler than glymphatic transport. It could be useful to simulate extreme cases which may be unrealistic for the brain but are simple enough to allow clear scientific intuition (e.g. large-porosity limits or cases with transport among compartments happening much faster or much slower). The authors must also consider how sensitive their key conclusions are to the particular parameter values chosen.

I have further comments about the manuscript in its current form, but explicating them here and now would be a poor use of time for everyone involved, because any further consideration of this manuscript for publication should first require the validations described above. Further consideration should also require improving the figure quality; Fig. 6 lacks labels altogether and is therefore unintelligible, whereas other figures are produced at resolution so low that they are difficult to read.

6. PLOS authors have the option to publish the peer review history of their article (what does this mean?). If published, this will include your full peer review and any attached files.

Reviewer #1: No

Reviewer #2: No

Reviewer #3: No

Reviewer #4: **Yes: **Douglas H Kelley

---

## [Author Response · Author response to Decision Letter 0]

16 Dec 2022

Authors’ response to reviewers 

Multi-compartmental model of glymphatic clearance of solutes in brain tissue

Alexandre Poulain, Jørgen Riseth, Vegard Vinje

PLOS one

December 14th 2022

We thank the reviewers for their feedback and constructive comments. Our detailed response follows below. The reviewers’ comments are included in italics, while our responses follow in roman text. Also, we refer to the file diff.pdf, a version of the manuscript indicating the changes made from the previous version. 

Reviewer #1: 

This interesting paper presents a discretized, “compartment” model of the so-called glymphatic system for brain clearance, consisting of advection and diffusion occurring in CSF and ISF in various parts of the rat brain. This is an ambitious effort to address an important feature of brain physiology and health. The model seems to be constructed with considerable care, and the results are of considerable interest. I recommend that the paper be published, but I have several questions and suggestions for the authors to consider.

Comment: We thank the reviewer for these encouraging comments. 

General comments:

1 . The model is applied to the clearance of a finite amount of tracer injected at an initial time, treated as an initial value problem in which the total amount of tracer in the brain decreases with time. On the other hand, the glymphatic system operates naturally to clear metabolic waste molecules that are produced continually in the brain, and this production and clearance are in balance and produce a steady state. Can the authors relate their model to such a steady-state clearance of metabolic waste?

Comment: We agree that the model could be useful for cases with continuous production and clearance. However, if we were to simulate e.g. amyloid-beta production and clearance, additional parameters would be needed, including convective and diffusive transport over the blood-brain barrier. As the present manuscript already contains several parameters that are to some extent unknown, we recommend that this extension of the model is postponed for future work. 

Action: We have put a remark in the limitations section, mentioning how this extension of the model can be used. 

The authors have carefully chosen numerical values of all of the parameters in the model, based on the best experimental data. However, some of these parameter values are subject to considerable uncertainly. It would be useful to point out the more important uncertainties. It would be quite useful to carry out a sensitivity analysis on the model, identifying the parameters for which the model is most sensitive to uncertainty in their values. An example of such a sensitivity analysis is that of Boster et al. 2022 (J. Roy Soc. Interface 19: 20220257), who performed a sensitivity analysis of the hydraulic network model of the glymphatic system described in Tithof et al. 2022 (your reference 72).

Comment: We agree with the reviewer that a sensitivity analysis is needed and have performed sensitivity analysis with variations in 8 parameters that we believe are uncertain and/or important for this problem. 

Action: We have added the precise description of the sensitivity analysis under methods and the corresponding results in the results section. Figures produced in the sensitivity analysis have been put in the supplementary material. 

Specific comments:

Abstract: Don’t capitalize glymphatic or biology.

Comment/Action: Corrected.

Page 2, par. 1: Reference 60 also argues that transport in the ECS is not by diffusion alone. Further theoretical evidence in favor of flow of ISF in the ECS, based on the sleep/wake variation in clearance, is given in Thomas 2022, FBCNS 19:30.

Comment: Reference 60 is the paper ”Analysis of convective and diffusive transport in the brain interstitium by Ray et al. “. We do, however, state that “In a recent study Ray et al. [60] concluded that transport of large molecules will be dominated by convection given expected ECS flow rates as reported in [27]..., which we understand to address the first comment. 

Action: Added reference to Thomas 2022 among the references who argue for convective mechanisms, and again under the effect of sleep under section 2.4.

Page 2, first two paragraphs: Fluid dynamic models, and in particular, previous compartment models, of CSF flow and transport are discussed in a very recent review article, Kelley and Thomas 2023, Annual Review of Fluid Dynamics, pre-publication version available online.

Comment/Action: Added a reference to the suggested review article.

Page 3, Remark 1: The correct term here is “homogeneous”, not “isotropic”. Isotropic means independent of direction, and you have already assumed isotropy by assume scalar, rather than tensor, permeability, etc.

Comment/Action: Corrected.

Page 7, section 2.3.2, Porosity coefficients: For the mouse brain, although the porosity of PVSs around penetrating arteries and arterioles is unknown, the PVSs around pial arteries have been shown to be essentially open spaces (Min Rivas et al. 2020, J. Roy. Soc. Interface, 17: 20200593), i.e., porosity = 1. Assuming this is also true in the rat brain, can you account for this in your model?

Comment: Our model results from a homogenization procedure starting from a microscopic description of the flow in each compartment. In our model, the PVSs themselves are assumed to be open spaces i.e. porosity = 1, but the open spaces do not occupy the entire geometry. Hence after homogenization, the porosity is not 1. Having a detailed PVS geometry with porosity = 1 requires detailed 3D-1D modelling, which is not the scope of the present study. Darcy’s law (macroscopic level) for the fluid velocity is obtained from homogenization of Stokes flow (microscopic description). See Hornung, Ulrich. Homogenization and porous media. Vol. 6. Springer Science & Business Media, 1996. 

Page 10, Remark 3: “precised”? Do you mean “specified”?

Comment/Action: corrected.

Page 10, The effect of sleep: The effect of sleep/wake variations in permeability is discussed in Thomas 2022, cited above.

Comment/Action: We added the reference proposed, under the effect of sleep (section 2.4). 

Page 13, Table 3 and text: Here you might want to compare with the flow speeds in pial PVSs in the mouse brain, measured by particle tracking in Mestre et al. 2018 (your reference 47). They are comparable to your values.

Comment/Action: We have added a few sentences comparing our results and Mestre et al (2018) results (in paragraph “Baseline parameter values for the multi-compartment model corresponds to the awake state” in our “Discussion” section). 

Page 15, section 3.3.1: Here, and perhaps elsewhere, you might want to point out the relative importance of advection and diffusion in the ECS in your various scenarios, best expressed in terms of the Peclet number.

Comment/Action: The Peclet number is now mentioned in Section 3.3.1, and is revisited in the discussion.

Reviewer #2: 

This work develops a mathematical multi compartment framework for modeling solute clearance from the brain by the convective and diffusion flow through the perivascular and extracellular, (and vascular) spaces. The paper is well written and straightforward to follow. The authors did an excellent job to include many physiological parameters into the modelling. However, I have two recommendations to improve the paper.

Comment: We thank the reviewer for constructive feedback and for careful consideration of the manuscript.

1) The results of this kind of modelling are very dependent on the prior knowledge of the model’s parameters. As appreciated by the authors, the experimental driven physiological parameters used in this model (Tables 1 and 2) are not precise and can vary considerably from one experiment to another. Moreover, even changing the structural parameters of the modeling (like the mesh size, number of compartments, etc.) can change the results [As studied in the results section]. Therefore, I would recommend doing a sensitivity analysis (although the authors mentioned this in their future plans, page 19) to study the sensitivity of the results to the input parameter variations. This may help researchers to know which parameter has a more dominant effect on the results.

Comment: We agree with the reviewer that a sensitivity analysis is needed and have performed sensitivity analysis with variations in 8 parameters that we believe are uncertain and/or important for this problem. 

Action: We have added the precise description of the sensitivity analysis under methods and the corresponding results in the results section. Figures produced in the sensitivity analysis have been put in the supplementary material. 

2) I would like the authors to discuss (in one or two paragraphs) about using this modelling in estimating physiological glymphatic parameters from dynamic contrast MRI measurements, like previous works such as (Lee et al.,J. Neurosci, 2015) and (Davoodi-Bojd et al., Neuroimage, 2019). Do the authors think their modelling can be used in a backward manner?

Comment: Even though we did not work in this direction yet, we believe that our mathematical model could be used in the future to estimate relevant parameters such as permeability or diffusion coefficients in the different compartments we represent. However, at least two major difficulties arise for the backward method:

Our model is composed of two coupled partial differential equations involving many parameters. Optimization techniques can be used to find the correct value such as to minimise the error between the output of our simulations and experiments (or MR images). Due to the great number of parameters the dimension of the optimization problem would be very large and, hence, computationally expensive. We are aware of recent developments in the field of PDE constrained optimization and we strongly believe that some optimization techniques could be used to find new results about the parameters of our model.

The second difficulty concerns the experiments that we can use to perform our inverse problem. Indeed, we do not know yet if the data that is accessible for the moment is sufficient to state an inverse problem.

Action: We added a paragraph about how inverse problems could be relevant in the limitations and future work section.

Minor comments:

1- The authors should provide a rational why they chose 14C-insulin for their modelling.

Comment/Action: Included two sentences (last paragraph of the introduction) describing the reasoning behind the choice of molecules.

2- Page 4, “where |Ω| =R Ω 1 dx = 2313 mm3 is the brain volume”. Please put a reference for the value of the brain volume.

Comment/Action: The volume is computed from our numerical mesh. An indication has been added in the manuscript after the value.

3- Page 5, please explain why the tracer distribution in the initial condition is assumed Gaussian? This implies there is no spatial restriction.

Comment/Action: Added a few sentences regarding the mass distribution of a Gaussian stating that the values are close to zero and negligible, as you go beyond a short distance surrounding the injection centre (96% of injected tracers are located within 2 mm from the injection center). In addition, we added a comment on how the projection of the initial condition onto the modelling space ensures that the boundary conditions are zero. 

4- Page 5, Boundary condition. It is known that the blood flow (and hence pressure) is changed during heart cycle, which cause vascular pulsation. Although it is hard to include this parameter in the modelling, but I would like the authors to elaborate on this parameter.

Comment: This information could be taken into account in our model assuming a periodic time dependent boundary condition for the fluid pressure equations. However, this would lead us to consider the non-stationary form of the pressure equation (that includes the time derivative of the pressure in the different compartments) and, hence, would change both the model and the numerical method (the pressure equation would need to be solved at each time step increasing drastically the computational cost). Furthermore, it would be necessary to refine the time step such as to capture the modification due to cardiac cycle (leading again to an increase of computational cost). This is why for the moment we did not consider this effect. However, It would be interesting to evaluate this effect in the future and compare it to the averaged effect we considered in the present work.

Action: TODO: added one/two sentence(s) in limitations that we did not include arterial pulsations. 

5- Page 6, “A decrease of molecules within the brain, corresponds to an increase of concentration in the SAS, and vice-versa.”. First, I assume “molecules” means the solute molecules (14C-Insulin), right? Secondly, I don’t understand the term “vice-versa” here. Does this mean solute molecules return back to the brain?

Comment: The reviewer is correct that overall, solutes do not return to the brain. However, at some local regions, the boundary concentration (homogeneous in space) is higher than the concentration within the brain tissue, which may cause solutes to return to the brain over a short period of time. Therefore, the term vice-versa refers to the back-and-forth communication between the brain and SAS. 

6- In the text there are several places the authors mentioned “molecules”. Please clarify if it refers to 14C-insulin.

Comment/Action: This has been modified in the manuscript, and we now use C14-inulin consistently.

Reviewer #3: 

Overall this is a well-produced and well-written manuscript that presents a new model of inter- and intra-compartmental solute transport and flows within and between the CSF and interstitial fluid of the brain, including effects of both diffusion and convection, as well as the porosity of these spaces. It could provide a very good tool for studying and evaluating aspects of the proposed glymphatics system and their feasibility, with one of the most interesting findings being how big an effect fluid leakage from larger blood vessels could have on the clearance of solutes (here 14C-inulin) from the brain. The manuscript is also clear with respect to data availability (e.g., code and meshes) and ethical considerations (where applicable, since it is a model study). Appropriate mesh and time step analyses have been performed. I do have some minor comments that I think would improve the manuscript.

Comment: We thank the reviewer for the careful reading and the encouraging words. 

Minor comments:

1) It is a bit unclear from figure 1 how flows are limited. In the figure description it says that fluid movement occurs along the red arrows, but couldn’t the arrows from the PVS to the ECS be bidirectional? Especially for the final simulation, where PVS flow directions change?

Comment: The direction of the arrows is decided based on our intuition on how fluid will flow in the models. In the 4-compartment model we decided the boundary conditions such that some flow directions are known a priori (e.g. the flow between arterial PVS and ECS). However, in the 7-compartment model, more complex relationships between compartments are considered. Therefore, we show potential flow in both directions where we are unsure of the outcome. In the final simulations the reviewer refers to, the PVS flow direction has changed, indicated by a double arrows between arterial PVS and all other extra-vascular compartments. 

Action: The figure has been updated to include the SAS. 

2) I also think that, in the beginning, it is a bit unclear that the SAS is handled through boundary conditions, as it does not seem to be included explicitly as a “compartment”. This should be clarified or emphasized more, early on. Is it possible to complement figure 1 with a more complete view of the entire model?

Comment/Action: The figure has been changed to include the SAS and communication with the other compartments.

3) Some single data values lack references (or the references are unclear). I am thinking of the p_pv, p_pvspial pressures for example, otherwise state that (if and how) they are calculated. I am also wondering about the ICP and ISF pressure references, since they are far apart in time and in different groups of rats. How representative are they? Are there no simultaneous measurements of ISF pressure and “ICP” in rats published? Here a sensitivity analysis would be appropriate to mention. Sensitivity analyses are mentioned in the limitations. Could they not be included in the current manuscript without problem?

Comment: We agree with the reviewer that some data values may lack references or be uncertain to some extent. For the boundary conditions, p_pa is the upper bound of the measured ICP in the reference [79], while p_pv is the lower bound in the same work. It is hard to know exactly how representative the measured pressure is. Therefore, we also agree with the reviewer that a sensitivity analysis is needed and we have performed sensitivity analysis with variations in 8 parameters that we believe are uncertain and/or important for this problem. One of these parameters include the boundary pressure from the SAS.

Action: We provided references for the parameter values requested by the reviewer. We have added the precise description of the sensitivity analysis under methods and the corresponding results in the results section. Figures produced in the sensitivity analysis have been put in the supplementary material. 

4) There are some small typos and grammatical errors here and there that can be adjusted (the manuscript lacks some final polish).

Comment/Action: We agree with the reviewer that this part can be improved. We went through the paper to look for grammatical errors and typos.

5) Figure 6 is missing its labels

Comment/Action: Thank you for pointing this out. The error occurred due to a mistake in the conversion between file formats to generate the separate figure files. We now made sure that the attached figures are as intended. 

6) The description of Figure 1 includes AEF and BBB but no such labels are seen in the figure.

Comment/Action: We had the same issue as for comment 5). This is now fixed.

7) Write explicitly that you are using the Laplace operator in the RHS of eq. (1) (diffusion equation. Most readers will know what it is but some may read it as a delta (gradient and divergence cannot be mistaken however).

Comment/Action: The definition of the Laplace operator has been added after Eq. 1.

8) Similarly, I think that e.g. the Darcy’s law should be referenced before equation 2 to add a pedagogical extra step to improve readability making it easier for any reader to assimilate the equations. At the very least a bit more info on where the equations come from would be an improvement.

Comment/Action: Precise definition and reference to the original work of Darcy has been added.

9) On page 8 the pressure drop from arteries to capillaries (and from capillaries to veins) are not the same as on page 22 (and the indexes are shifted, and the Pc-v is named Pa-c on page 22).

Comment: We thank the reviewer for pointing out this inconsistency. There was a typo in the manuscript. This is now fixed. 

Action: The parameters have been corrected and are now the same in the main body and the appendix.

10) On page 15, for the simulation where ECS porosity is increased, the velocity field seems directed towards the ventricles, as the authors state. Do the authors find this reasonable/physiological? It is mentioned but not really discussed in the discussion section (and that increases in the porosity of ECS seem to act in the opposite direction to increases in PVS. Reasonable?). The results are very useful regardless, however some additional comments would be appropriate.

Comment: Thank you for making us aware of this error. The wording in the submitted manuscript was misleading/wrong, as the velocity field is actually directed inwards away from all surfaces, including the ventricles. The main point of the statement was that the velocity field is directed away from the regions responsible for solute clearance, hence inhibiting the clearance. 

Action: The wording has been changed to make this point clear. 

11) Also, I may have missed it, but are the calculated flow velocities in the PVS compared or validated against other estimations in the literature? (The comparisons of clearance to that in the literature are good, however)

Comment: Flow velocities in PVS have to our knowledge only been measured on (or very close) to the brain surface. Within brain tissue, there are some estimates on the “bulk flow”, taking into account an averaged flow over all compartments. 

Action: We have added further comparisons with fluid flow velocities. We now compare with our model results both in arterial PVS (Mestre et al. (2018), Bedussi et al. (2018)) and in the ECS (Ray et al. (2019), Nicholson (1981), Cserr et al. (1981, 1977) Abbott (2004). 

12) I have a hard time understanding the first paragraph of the limitation section where the authors bring up the 1D-3D approach. Could the authors clarify the main differences of this model compared to the current one and especially how it is more computationally costly?

Comment: 1D-3D models require mesh resolution on the order of the vessels. The smallest vessels in the brain are on the order of a few μm, while our mesh is sufficiently fine with a resolution greater than 150 μm (Table 8). Therefore, resolving all vessels would require 1) a coupling between the 1D and 3D spaces and 2) a substantial refinement of the mesh, hence drastically increasing the computational cost. 

Action: Elaborated a bit further on 1D-3D models, and pointed out how these models typically require high mesh resolution with cell sizes on the scale of blood vessel radii (with a reference backing up the claim).

Reviewer #4:

 This manuscript presents numerical simulations of glymphatic clearance of brain solutes, that is, clearance by the combined effects of diffusion and advection by flowing cerebrospinal fluid. A novel numerical model is described, in which the authors treated the brain as a set of interpenetrating, communicating fluid compartments: extracellular space, periarterial space, perivenous space, pericapillary space, arteries, veins, and capillaries. In the model, all compartments co-exist throughout the brain without spatial separation, and all are treated as porous media, such that effects of small-scale channel geometries are estimated by bulk porosity and permeability coefficients. Transport of fluid and solutes among compartments is presumed to occur through membranes. In some simulations, blood compartments are not explicitly considered. The authors present pressure fields and spatiotemporal solute evolution as predicted by the simulations. From the results, the authors conclude that clearance rates depend on the size of the sample considered, that solute boundary conditions in the subarachnoid space surrounding the brain affect predicted clearance rates, that the chosen parameter values seem to correspond to transport during wakefulness as observed by Xie et al. (Sleep drives metabolite clearance from the adult brain, Science, 2013), that increasing the porosity (volume) of extracellular space slows clearance, and that allowing fluid outflow from blood compartments causes fast clearance because fluid is driven to flow toward the brain surface in the extracellular and perivascular spaces.

This numerical model is quite different from others used previously to study glymphatic clearance and thus has potential to bring new insights. Unfortunately, the authors have not validated the model. Though grid convergence studies and time step size effects were considered for a few cases, those necessary steps are not sufficient to show that a model as complicated as this one is valid and trustworthy. I have less concern about the numerical methods than about the dozens of biophysical values which must be chosen as input parameters: porosities, permeabilities, pressures, and more. Few have been measured directly. The chosen values often depend on complicated reasoning involving many simplifying assumptions (see [Supplementary-material pone.0280501.s001]). Though that reasoning usually seems credible, the authors must go further by demonstrating that the values and the model accurately reproduce real, established biology. The model must be used to solve a problem whose answer is well-known. The problem should involve multiple compartments, porous media, and solute transport by both diffusion and advection. Experimental measurements should be used for comparison; solutions from prior models might also be used. Simulating cases where closed-form analytic solutions are available would also help, though they are likely to be drastically simpler than glymphatic transport. It could be useful to simulate extreme cases which may be unrealistic for the brain but are simple enough to allow clear scientific intuition (e.g. large-porosity limits or cases with transport among compartments happening much faster or much slower). The authors must also consider how sensitive their key conclusions are to the particular parameter values chosen.

I have further comments about the manuscript in its current form, but explicating them here and now would be a poor use of time for everyone involved, because any further consideration of this manuscript for publication should first require the validations described above. Further consideration should also require improving the figure quality; Fig. 6 lacks labels altogether and is therefore unintelligible, whereas other figures are produced at resolution so low that they are difficult to read. 

Comment: We thank the reviewer for careful feedback. We apologise for the figure resolution and labels lacking, as there were some conversion issues in the submission pipeline. We have made sure the figures are of high quality in the new manuscript. 

Verification of model: 

Comment/Action: A convergence study using the method of manufactured solution has been performed, which revealed expected convergence of the numerical method. 

Validation of the model: 

Comment: We acknowledge that all aspects of the model can not be validated against experimental data. For instance, no direct data exist on molecular transport along PVS within the brain. We still believe that our model can provide answers to relevant biological questions, given that we perform the sensitivity analysis and comment upon the results and the parameter range. Our model reproduces to some degree results found in experimental studies. PVS flow speed matches well with results obtained by Mestre et al. (2018) and Bedussi et al. (2018). However, results from these studies are only reported on the brain surface, and hence the comparison with our model is only made on the brain surface. For PVS flow speeds within the brain, there are to the authors knowledge, no experimental studies that have addressed this, and there is even a debate on whether PVS deeper in brain tissue actually exists (Hannocks et al., JCBFM 2018). For solute transport and clearance, we note that clearance is qualitatively in line with results obtained by Xie et al. (2013). However, as shown in Figure 4a, the comparison depends on the size of the measurement sample from our model. From our understanding of the experimental procedures, the results from Xie et al. on clearance needs to be compared to our results taking the entire mouse geometry as the measurement sample. In that case the results from the 4-compartment model suggest slower clearance in the sleeping versus the awake state, suggesting that an increase in the extracellular space alone is not sufficient to explain the observed difference in clearance during sleep versus awake. 

Action: We have added rate constants and compared with Xie et al. (2013) for both awake and sleep.

Sensitivity analysis: 

Comment: We are thankful that the reviewer found our work to establish baseline parameters credible as we put a lot of work into this part. However, we agree with the reviewer that a sensitivity analysis is needed and have performed sensitivity analysis with variations in 8 parameters that we believe are uncertain and/or important for this problem. We find that some model parameters change the resulting clearance, while some do not. In particular, the convective coupling between arteries and arterial PVS is the parameter that may influence the results the most. This was not surprising as this is essentially a continuous bridge between the 4-compartment and 7-compartment model, and we find that increasing this parameter further increases clearance in the 7-compartment model. In addition, we find that a change in the diffusion coefficient highly affects clearance in the 4-compartment model, and to some degree in the 7-compartment model. This suggests that the Peclet numbers are in the transition regime and that the role of diffusion versus convection will be hard to determine exactly.

Action: We have added the precise description of the sensitivity analysis under methods and the corresponding results in the results section. Figures produced in the sensitivity analysis have been put in the supplementary material.

---

## [Decision Letter · Decision Letter 1]

3 Jan 2023

Multi-compartmental model of glymphatic clearance of solutes in brain tissue

PONE-D-22-27008R1

Dear Dr. Vinje,

We’re pleased to inform you that your manuscript has been judged scientifically suitable for publication and will be formally accepted for publication once it meets all outstanding technical requirements.

Kind regards,

Quan Jiang, Ph,D.

Academic Editor

PLOS ONE

Additional Editor Comments (optional):

Reviewers' comments:

Reviewer's Responses to Questions

**Comments to the Author**

1. If the authors have adequately addressed your comments raised in a previous round of review and you feel that this manuscript is now acceptable for publication, you may indicate that here to bypass the “Comments to the Author” section, enter your conflict of interest statement in the “Confidential to Editor” section, and submit your "Accept" recommendation.

Reviewer #1: All comments have been addressed

Reviewer #2: All comments have been addressed

Reviewer #3: All comments have been addressed

2. Is the manuscript technically sound, and do the data support the conclusions?

Reviewer #1: Yes

Reviewer #2: Yes

Reviewer #3: Yes

3. Has the statistical analysis been performed appropriately and rigorously? 

Reviewer #1: Yes

Reviewer #2: N/A

Reviewer #3: N/A

4. Have the authors made all data underlying the findings in their manuscript fully available?

Reviewer #1: Yes

Reviewer #2: Yes

Reviewer #3: Yes

5. Is the manuscript presented in an intelligible fashion and written in standard English?

Reviewer #1: Yes

Reviewer #2: Yes

Reviewer #3: Yes

6. Review Comments to the Author

Reviewer #1: The authors have responded positively to all of my suggestions. I was especially pleased to see that they performed a sensitivity analysis of their model for several important parameters.

Reviewer #2: (No Response)

Reviewer #3: The manuscript has improved substantially and my comments have been adressed with proper changes or clarifications. Many numbers were updated this time around, and it is important to double-check that they all are correct. Other than that, I have no further comments.

7. PLOS authors have the option to publish the peer review history of their article (what does this mean?). If published, this will include your full peer review and any attached files.

Reviewer #1: No

Reviewer #2: No

Reviewer #3: No

---

## [Editor Report · Acceptance letter]

11 Jan 2023

PONE-D-22-27008R1 

Multi-compartmental model of glymphatic clearance of solutes in brain tissue 

Dear Dr. Vinje:

I'm pleased to inform you that your manuscript has been deemed suitable for publication in PLOS ONE. Congratulations! Your manuscript is now with our production department. 

Kind regards, 

on behalf of

Dr. Quan Jiang 

Academic Editor

PLOS ONE